civil engineering/structural engineering

steel pipe, buried pipe, bending, soil movement, FEA numerical analysis

**Author for correspondence:**
Mohamed Almahakeri
e-mail: almah1my@cmich.edu

# Numerical techniques for design calculations of longitudinal bending in buried steel pipes subjected to lateral Earth movements

## Mohamed Almahakeri[1], Ian D. Moore[2] and Amir Fam[2]

[1]School of Engineering and Technology, Central Michigan University, Mount Pleasant, MI 48859, USA
[2]Department of Civil Engineering, Queen's University, Kingston, Ontario, Canada K7L 3N6

MA, 0000-0001-5188-5927; IDM, 0000-0003-2446-1891

This paper presents simplified finite-element analysis procedures based on geometrical nonlinearity and ductile Mohr–Coulomb–Davis plasticity for analysis of bending behaviour of steel pipes subjected to lateral soil loading. A simple, and easy to implement, user-defined subroutine to represent soil stiffness using the Janbu model is also presented and discussed. The development of a three-dimensional (3D) finite-element model is presented, and its evaluation against experimental measurements is discussed. Data are presented for different burial depths of the pipe, including soil loading on the pipe as well as 3D responses, longitudinal bending deflections and pressure distribution along the pipe. It was shown that numerical analyses which include soil modulus dependency on confining pressure lead to effective 3D calculations of pulling forces, bending moments along the pipeline and flexural deformations, based on measured soil parameters. The 3D analysis model requires the use of lower order (linear displacement) elements, which overestimated peak mobilized load. However, those 3D calculations effectively provided the progress of both the load–deflection and longitudinal bending response of the steel pipe at embedment ratios up to 5 where most energy pipelines are buried.

## 1. Introduction

Oil and gas transmission lines can cross zones of soil instability and may need to be designed to resist differential ground movements. Soil instabilities can result from natural phenomena such as soil creep, slope failures, landslides and seismic excitations in the

**Table 1.** Notation: the following symbols and abbreviations are used in this paper.

| | |
|---|---|
| $c$ | cohesion of the soil |
| $D$ | pipe diameter |
| $E_s$ | soil modulus of elasticity |
| $H$ | burial depth to the pipe spring line |
| $K_o$ | the coefficient of lateral Earth pressure at rest |
| $L$ | pipe length |
| $N_{qh}$ | horizontal bearing capacity factor for sand |
| $P_u$ | peak soil resistance load to lateral pipe movement |
| $P_{resid}$ | residual soil resistance load to lateral pipe movement |
| $P_o$ | atmospheric pressure at sea level |
| $T$ | pipe wall thickness |
| $Y_u$ | rigid pipe displacement at peak pulling load |
| $Y_{u,e}$ | flexible pipe end displacement at peak pulling load |
| $Y_{u,m}$ | flexible pipe mid-span displacement at peak pulling load |
| $\alpha$ | plastic strain magnitude |
| $\delta$ | bending deflection at mid-span of the pipe |
| $\phi_p$ | peak friction angle of sand |
| $\phi_{resid}$ | residual friction angle of sand |
| $\gamma$ | bulk unit weight of soil |
| $\gamma'$ | effective unit weight of soil |
| $\mu$ | contact friction coefficient between soil and pipe surfaces |
| $\nu$ | Poisson ratio |
| $\sigma_1$ | maximum effective principal stress |
| $\sigma_3$ | minimum effective principal stress |
| $\theta_{int}$ | pipe–soil contact interface friction angle |
| $\tau_{crit}$ | pipe–soil contact interface critical shear stress |
| $\psi$ | dilation angle of sand |

proximity of tectonic faults. Human activities such as excavation, tunnelling, directional drilling and pipe bursting can also lead to lateral soil movements against pipes. The pipe can actually be dragged through the soil mass to a point where exceedance of stresses owing to bending can jeopardize the structural integrity of the pipe. Moreover, even if the structural safety is guaranteed through sizing of a sufficiently flexible pipe, serviceability design limits may still be exceeded (e.g. local ovalization or wrinkling).

With the ever increasing construction of energy pipeline projects around the world, there is a growing need for more in-depth understanding of three-dimensional (3D) effects on deformable pipes. Full-scale testing and site investigation are two very accurate methods to achieve this objective. However, considering the high cost associated with such studies, it is generally more economical and less time-consuming to perform numerical simulations. This method permits broader investigation of the pipe–soil interaction by performing parametric analyses on material properties (both for the pipe and the soil), geometrical configurations (pipe size and embedment ratios) and loading conditions (combined loading and pipe pressurization). On the other hand, data obtained from field surveys or from laboratory tests can be useful for calibrating or validating such models.

In pipe–soil interaction problems, it is known that the restraint loads acting on the pipeline are governed by the soil stress–strain constitutive law. Over the years, numerous researchers using finite-element simulations have attempted to capture the influence of different soil parameters on the load-carrying capacity. Popescu *et al.* [1] presented a two-dimensional (2D) numerical study on pipes translating through the ground. The developed model is based on non-associated Mohr–Coulomb material model, with hardening of soil parameters cohesion, $c$, and friction angle, $\phi$, defined as functions of plastic strain

magnitude, $\alpha$ (all symbols used in this article are included in table 1). Mahdavi et al. [2] complemented the numerical investigation of Popescu et al. [1] by examining the effect of parameters such as nominal pipe diameter, pipe diameter to wall thickness ratio ($D/t$), steel grade, internal pressure, axial force and embedment ratio ($H/D$) on local buckling response. Guo & Stolle [3] conducted a numerical parametric study on the effect of embedment ratio, soil parameters and scale effect on soil capacity for a 2D numerical model. Phillips et al. [4] examined the impact of combined axial load and lateral pipe–soil interaction, while Yimsiri et al. [5] studied pipe–soil interaction for deep embedment conditions ($H/D = 11.5$ to $100$ examined). Takada et al. [6], Abdoun et al. [7], O'Rourke [8], Vazouras et al. [9,10], Saiyar et al. [11] and Robert et al. [12] conducted numerical studies focusing on pipelines crossing tectonic faults. Robert et al. [13] performed 2D finite model analyses to predict lateral loads on pipelines in dry and unsaturated sandy soils and were compared with full-scale tests. Cheong et al. [14] investigated the pipe–soil interaction owing to lateral loading, but with emphasis on bent pipes. While most of the computational work adopted implicit finite-element analysis (FEA), different numerical approaches have been used as well. Roy et al. [15] performed simulations using the explicit solver to capture more accurately the strength degradation in shear bands compared to the implicit solver. Other different numerical simulation studies include the work of Calvetti et al. [16], employed the distinct element method, and Karimian [17], used the finite difference method. In the present study, simple 2D and 3D finite-element models employing stress-dependent stiffness of the soil using the Janbu model subroutine are presented and discussed.

Most guidelines for the design of buried pipelines (e.g. [18–20]) employ a structural representation of the pipe–soil interaction, where a series of independent nonlinear structural springs in the axial, lateral and vertical directions are used to represent the soil support, while the pipeline is simulated using beam elements. One of the most widely used design approaches is the American Society of Civil Engineers (ASCE) guidelines [20], according to which the lateral soil reaction on a pipe is defined as follows:

$$P_u = N_{qh}.\gamma'.H.D, \tag{1.1}$$

where $P_u$ is the peak load per unit length; $\gamma'$ is the effective unit weight of soil; $H$ is the burial depth to the pipe spring line; $D$ is the outer pipe diameter; $N_{qh}$ is the horizontal bearing capacity for sand.

$N_{qh}$ is a dimensionless parameter that depends on soil friction and embedment ratio ($H/D$). The ASCE guidelines [20] provide different charts to determine this parameter. One chart is based on the method developed by Hansen [21], and the other is based on the research work of Trautmann & O'Rourke [22]. While such simplification provides a convenient analysis tool that can be implemented in the design, springs do not model the through-soil interaction (connecting points as well as connecting the vertical and horizontal resistance of the soil), which limits the prediction accuracy (e.g. [2,4,23]).

The objectives of the present work are: (i) to develop a simple 3D modelling technique to estimate the load–deflection response of pipes moving laterally through the ground, based on independent parameters for the pipe and soil; (ii) to assess the performance of the model through comparisons against the most commonly used 2D modelling which enables the use of higher order elements, and then compare both anayses with measured behaviour; and (iii) to use the 3D analysis to examine pipe bending, pressure distribution along the pipe and also to develop calculation methods that address challenges with the low-order elements that must generally be used for 3D soil–pipe interaction analysis (where failure loads are usually overestimated). These computational tools will then be available to assess a variety of buried pipe bending problems of interest to the oil and gas pipeline industry.

## 2. Testing programme

A series of physical tests were performed at the geotechnical laboratory testing facility at Queen's University [24]. The test pit and test set-up are shown in figure 1. A total of nine tests were conducted to examine effects of burial depth (ratios of 3, 5 and 7), boundary effects, internal friction of the pulling mechanism (interactions between the pulling system and the soil) and repeatability of testing on the load–displacement curve. In each test, a mild-steel pipe of 102 mm diameter was buried in uniformly graded dense Olivine sand [24], which was then pulled laterally using a hydraulic actuator. The actuator is attached to one side of a spreader beam, where two steel cables (encased in PVC tubes to minimize contact friction with the surrounding soil) are attached to the other side of the spreader beam and run through the soil mass connecting to the pipe ends. Pulling was applied at a fixed rate of 5 mm min$^{-1}$. Soil and pipe parameters used in the analysis are given in tables 2 and 3, respectively. Details of the soil characterization tests including grain size analysis and triaxial tests are discussed by Almahakeri [25]. During testing, pipe deflection was measured at five sections along the pipe using

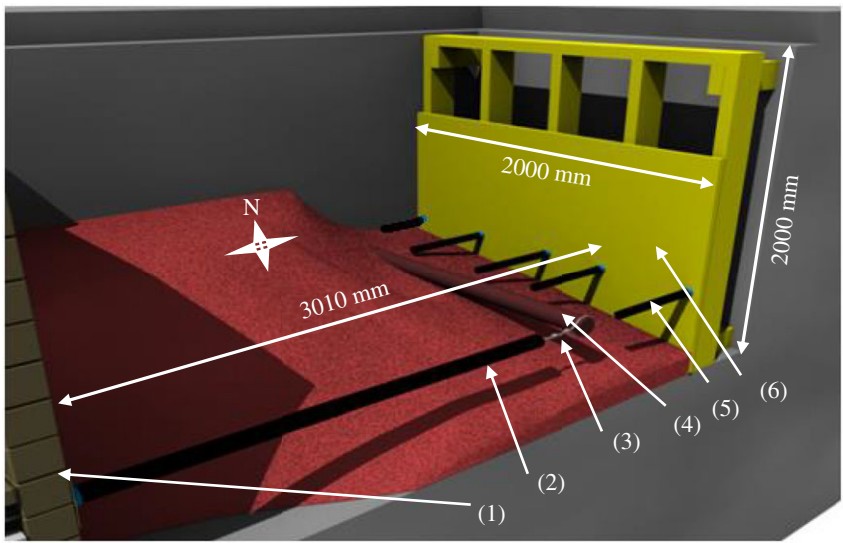

**Figure 1.** Test set-up schematic (3D view). 1, front end wall (retaining wall); 2, PVC encasing tube (see figure 2 for details); 3, pulling cable; 4, test pipe; 5, string potentiometers encasing tubes; 6, rear end wall.

**Table 2.** Soil parameters.

| parameter | value |
|---|---|
| modulus of elasticity ($E$), kPa | see figure 3 |
| Poisson ratio ($\nu$) | 0.3 |
| bulk unit weight ($\gamma$), kN m$^{-3}$ | 15.1 |
| friction angle ($\phi$) (peak/residual) | 53°/45° |
| angle of dilation ($\psi$) | 16° |
| cohesion ($c$), kPa | 1[a] |

[a]A non-zero value must be used for the numerical analysis.

**Table 3.** Pipe parameters.

| parameter | value |
|---|---|
| diameter ($D$), mm | 102 (4 inch) |
| length ($L$), mm | 1083 (6 ft.) |
| wall thickness ($t$), mm | 2.7[a] |
| modulus of elasticity (GPa) | 200 |
| material specifications | ASTM A5.13 |
| plasticity definition | table 4 |

[a]Measured (different from the 2.1 mm (0.083″) value from ASTM A5.13 and the supplier).

string potentiometers and strain gauges were fixed to the outer surface of the pipe (see figure 2 for a layout of the pipe instrumentation). Because the test with burial depth of $H/D = 3$ was repeated four times [24] to check for repeatability of results, the test with close to the average results (test no. S4A) relative to the other three tests has been selected for comparison with the numerical calculations in this paper (variations in loads at specific levels of lateral displacement were within 10% and 6% of the average peak and residual pulling loads, respectively).

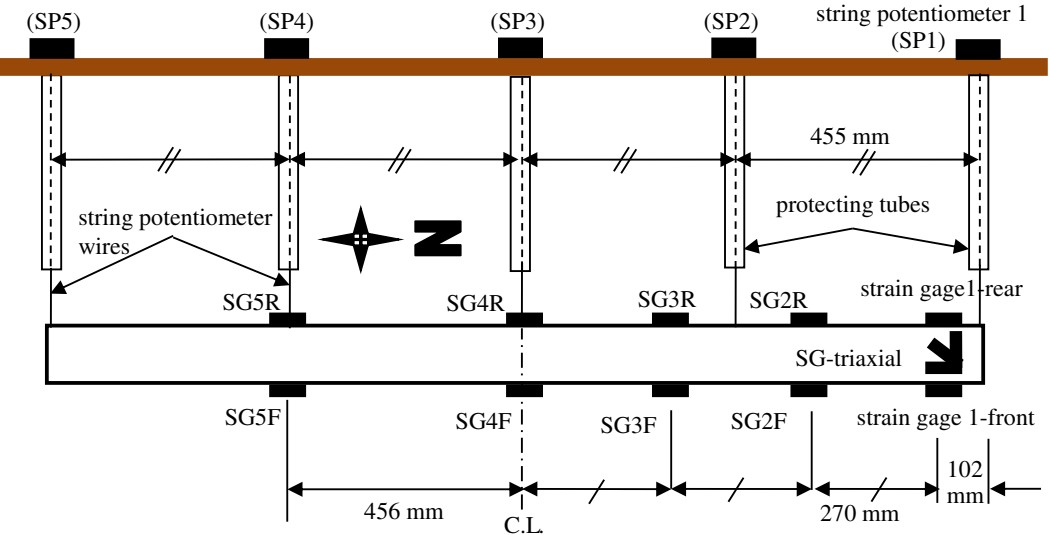

**Figure 2.** Pipe instrumentation layout (top view) (pulling cables not shown).

# 3. Finite-element model

FEA was conducted employing the commercial software ABAQUS/STANDARD, which was used to develop both 2D and 3D models of the tests being studied. The aim of the 2D model was to establish a simple plane strain approximation of the pipe–soil interaction that can be used for comparison with 3D calculations and to assess the ability of these different approaches to capture the peak forces mobilized in the soil (i.e. the peak pulling forces). Both of the numerical models—without the subroutine—were initially built to simulate the experiments of Karimian [17] for validation and verification purposes. Those tests featured rigid buried pipe displaced laterally in dense uniform sand.

## 3.1. Constitutive model

Several constitutive models have been used in the literature to simulate soil behaviour in soil-structure interaction studies (e.g. Lade [26], the Nor-Sand model of Jefferies [27] and modified Mohr–Coulomb by Robert [28]). While models like these are designed to capture as many aspects of soil behaviour as possible, they often require the determination of more soil parameters than the basic ones, such as the friction angle ($\phi$), dilation angle ($\psi$), unit weight ($\gamma$) and modulus ($E_s$) which can be obtained using standard soil tests. Any special testing of soils to obtain other parameters may be complex, time-consuming and expensive. In engineering practice, designers and researchers generally prefer to use simple models with sufficient accuracy to capture the main features of soil behaviour relevant to the particular problem of interest.

In the current study, emphasis is given to the stress–strain behaviour up to the failure of the soil mass. The simple elastic-perfectly-plastic model with the non-associated Mohr–Coulomb failure criterion implemented in ABAQUS/STANDARD is therefore used to describe the behaviour of the soil. This model has been successfully used (with different modifications) in several analyses involving large displacements of pipe–soil interaction [1,3,23,29]. Elastic soil modulus, $E_s$, is defined as a power function of the confining stresses according to the model proposed by Janbu [30]:

$$\frac{E_s}{P_o} = K \left(\frac{\sigma_3}{P_o}\right)^n, \tag{3.1}$$

where $P_o$ is the atmospheric pressure at sea level (=101.3 kPa), and the Janbu model parameters, $n$ and $K$, are soil-specific parameters. The minimum effective principal stress, $\sigma_3$, can be calculated from the following relation:

$$\sigma_3 = K_o \cdot \sigma_1, \tag{3.2}$$

where $K_o$ is the coefficient of lateral earth pressure at rest and $\sigma_1$ is the maximum effective principal stress which is the overburden stress, a function of the soil depth $H$ as follows:

$$\sigma_1 = \gamma.H. \tag{3.3}$$

By substituting equations (3.2) and (3.3) into equation (3.1), the soil modulus profile $E_\mathrm{s}$ is

$$E_\mathrm{s} = P_\mathrm{o}.K.\left(\frac{K_\mathrm{o}.\gamma.H}{P_\mathrm{o}}\right)^n. \tag{3.4}$$

Parameters $n$ and $K$, for the Olivine test sand have been determined by Almahakeri *et al*. [24] to be 0.86 and 326, respectively.

The elastic modulus of the soil $E_\mathrm{s}$ has been shown to be a major factor affecting the force–displacement curve, and has also led to some convergence issues which required some localized calibration in other studies to overcome those issues [2]. Furthermore, for a given constant $E_\mathrm{s}$ value used in an analysis, the slope of the pulling force–pipe deflection curve is affected, which in turn has an impact on pipe bending as well [1], even if the peak pulling load is essentially the same in both cases.

Initial values of soil modulus distribution, $E_\mathrm{s\_init}$, dependent on depth were implemented in the current study using a user-defined subroutine built in ABAQUS called USDFLD (written in FORTRAN). It allows the user to define and apply predefined field variables at all the integration points of the material as a function of any of the material attributes, or output quantities available in the ABAQUS analysis output file (e.g. integration point position coordinates, strain, and stress values, or the current loading step).

First, such field functions are evaluated at some selected discrete points (in the current analysis, it would be evaluating the soil modulus of elasticity, $E_\mathrm{soil\_init}$, as a function of the vertical coordinate of the integration point), then, the actual value of modulus—to be used in the analysis—would be the linear interpolation between these predefined values. This is usually accomplished by writing the field function as a number of 'IF' statements in the body of the subroutine, each one is used to control a corresponding field variable (which is, in this case, the soil modulus). Each field variable has a corresponding value that is defined through the user interface of the software. Those values depend on the soil elastic parameters ($n$ and $K$) calculated using equation (3.4). It has been observed that by increasing the number of the pre-calculated points on the curve (and consequently, the number of the corresponding 'IF' statements), the analysis completion progress was gradually advancing and that the calculated load–displacement curve gets closer to the one obtained from the experimental results. Hence, an 'IF' statement was used to represent the soil modulus for each 0.001 inch of soil depth. A script of the subroutine script is provided in the electronic supplementary material, appendix SA.

An important numerical issue arises for the modulus value at the ground surface given by the Janbu model. Equation (3.4) indicates that the modulus approaches zero near the surface, which, as expected, causes numerical convergence problems. Hence, a minimum, non-zero value of the soil modulus was assigned to the top soil layer (layer thickness depends on the pipe burial depth ratio that is analysed). Figure 3 shows the modulus of elasticity distribution over the soil depth used for the 3D analyses. The distributions used in the 2D analyses were identical, except that the minimum modulus value possible to run an analysis was found to be 400 kPa. Concerning the cohesion, $c$, since the Mohr–Coulomb model implemented in ABAQUS requires non-zero cohesion to accommodate the shape of flow potential close to the apex of the model, a minimum value of 1 kPa was employed even though the sand was dry. This artificial cohesion was introduced to ensure the soil had a small, but non-zero, strength when in a state of zero confining stress. This is a common requirement reported in several previous studies [3,31,32].

The non-associated flow rule was used to simulate the dilatant behaviour of the soil with constant dilation angle of 16° [25]. While coarse grained soils often exhibit reductions in dilation angle with increases in plastic strain, neglecting of this effect was found to influence the load–displacement curve in the post-peak region. Hence, changes in dilation angle should not significantly affect the pre-peak results, and the added complexity of reducing the dilation angle at large values of shear strain is not required. A constant value for the friction angle of shearing resistance of the soil ($\phi$) was also used, to avoid issues associated with modelling of strain softening (mesh dependency, e.g. [33]). Both the peak friction angle ($\phi_\mathrm{p}$) and the constant volume friction angle ($\phi_\mathrm{resid}$) were used in separate analyses to examine the calculated soil behaviour for each of these limits. These choices were made because the current study focuses on responses up to peak lateral loads, and also because methods developed to capture post-peak behaviour for 2D plane strain problems (modelling of decreases in shear strength and dilation angle like that reported by Anastasopoulos *et al*. [34] as presented by O'Rourke [8]) have not been demonstrated for the 3D calculations needed to calculate the longitudinal pipe bending, where the emphasis is given in the current study. A sensitivity study on the effect of its

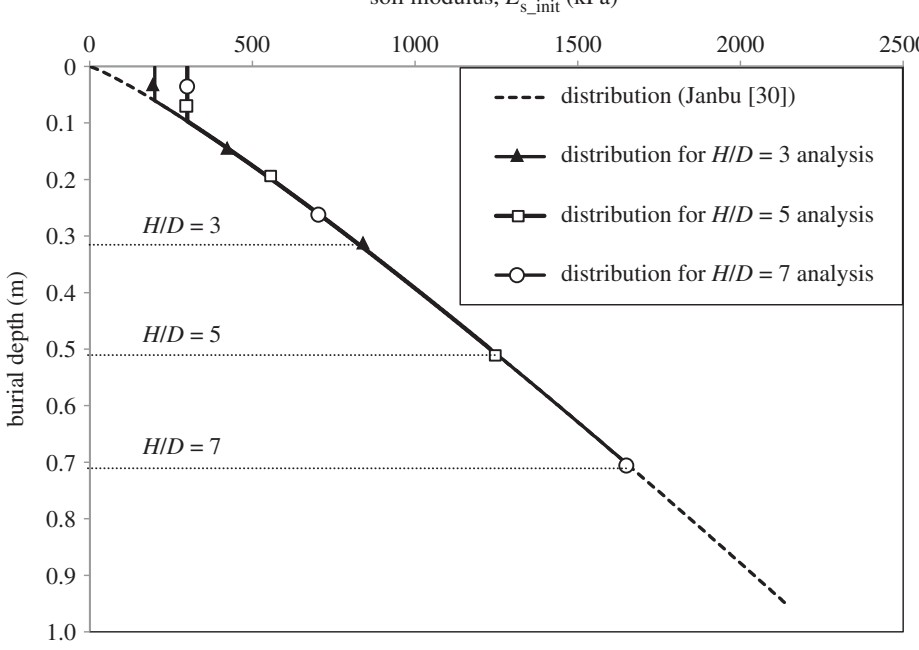

**Figure 3.** Initial soil modulus distribution over the burial depth (3D model).

**Table 4.** True stress – strain relationship for steel pipe material.

| true stress (MPa) | true plastic strain |
| --- | --- |
| 336.05 | 0 |
| 351.26 | 0.000519 |
| 377.13 | 0.00271 |
| 399.07 | 0.00744 |
| 455.55 | 0.043875 |
| 558.41 | 0.170275 |

variation showed almost no impact on the elastic portion of the load–displacement curve. For this reason, a fixed dilation angle is used.

For the pipe material, a piecewise elastoplastic response for the steel was modelled using the isotropic Von Mises plasticity model available in ABAQUS/STANDARD (in accordance with the properties provided in tables 3 and 4).

## 3.2. Element type selection and mesh refinement

For the 2D analysis, a plane strain, six-noded modified quadratic plane strain triangles (CPE6M) were used as the base modelling elements for both the soil and the pipe. This type of element is useful for modelling areas with irregular geometries and curvatures and their expected deformations. The analysis was also performed using the three-noded linear triangle (CPE3) and results are compared and discussed in the 'Results and discussion' section. The use of plane strain elements with reduced integration (CPE4R) led to severe penetration of the pipe's elements nodes into the soil elements, which in turn resulted in convergence difficulties during the analyses. To eliminate this effect, an analytical rigid body was used for the pipe part. While analyses started and progressed successfully using this approach, it was observed that excessive stiffening of the soil had an impact on the numerical stability. Hence, second-order triangular elements were adopted for the 2D analyses presented and discussed in this study. For the 3D model, analysis was attempted using the 20-node hexahedral, reduced integration element (C3D20R) for the soil, but owing to intrinsic convergence

issues with quadratic (higher order) elements in contact problems (as will be discussed subsequently), the element had to be substituted by eight-noded linear (lower order) hexahedral elements (C3D8). Because 'lower order' elements are overly stiff, extremely fine meshes are often required to obtain accurate results. For the pipe elements, four-noded reduced integration shell elements (S4R) were employed in the 3D model. This type of element account for finite membrane strains and arbitrarily large rotations and are therefore suitable for large strain problems like the one treated herein.

Mesh sensitivity analysis of the soil was performed for both the 2D and 3D numerical models. The obtained load–displacement curves for three different mesh refinements ('fine', 'coarse' and 'intermediate', as shown in figure 4) are illustrated in figure 5. Biased node seeding for the mesh elements of the soil was applied to permit finer mesh elements adjacent to the pipe–soil interface.

The automatic built-in mesh generation of the software was employed as the structured mesh configuration encountered numerical instability for the CPE6M elements (analysis aborted at 10.3% of completion). Comparison of the load–displacement curve between the two mesh configurations over the part of the curve where numerical convergence is met showed almost identical behaviour. The 'intermediate' mesh was selected in this study for both the 2D and 3D analyses (for an embedment ratio $H/D = 3$, element size ranges between 11 and 60 mm). Mesh sensitivity analysis on the load–displacement curve revealed almost identical behaviour up to the onset of plastic deformation of the soil for all examined mesh densities (2D and 3D), with minimal reduction in the level of strain hardening obtained with finer meshes at higher deformations (end displacement greater than 20 mm). On the other hand, increased mesh refinement leads to lower bending moment than coarser meshes (figure 5) with more pronounced rates than the load–displacement curve of the soil (12.5% reduction in peak bending moment between the coarse and intermediate meshes, and 8% reduction between the intermediate and fine meshes). However, the analysis time of the model using fine mesh was significantly longer than the one with the intermediate mesh. Based on these results, and in order to maintain a computationally efficient model, the mesh with intermediate refinement (figure 4e) was selected for the 3D analyses reported in what follows.

For the pipe, a fixed number of 40 elements were used in the 2D analyses to discretize the pipe over the circumference. For the 3D analysis, a total number of 28 elements were used around the pipe circumference (using biased seeding to increase refinement at the front spring line of the pipe), while 61 elements were used along the pipe length (one element per 15 mm of pipe length). Doubling the mesh refinement of the pipe mesh had no particular impact on the load–displacement curve and hence the initial mesh size was adopted for the analysis.

## 3.3. Contact interface modelling

A 'surface-to-surface' contact interface was used between the pipe and the soil surfaces. Moreover, a finite displacement contact interaction formulation was employed. This contact modelling approach is suitable for problems involving large sliding, separation and arbitrary rotation of the contact surfaces. Tangential and normal behaviour is defined between the two surfaces. For the normal interaction behaviour, a non-penetrating condition was defined (referred as 'hard' contact in ABAQUS), while for the tangential direction, the friction coefficient, $\mu$, between the pipe and the soil was introduced. Sliding occurs when the shear stress acting on the contact surface exceeds the critical shear stress, $\tau_{crit}$. The friction coefficient $\mu$ is calculated as the tangent of the interface friction angle, $\theta_{int}$, between the pipe and the test sand. This angle is usually determined by means of direct shear testing of the two surfaces. Owing to the absence of experimental data for $\theta_{int}$, the approach of Kulhawy et al. [35] was used, where the friction coefficient ranges from 0.5 to 0.7 of $\phi_p$. Because the tested pipe is new and relatively smooth, the lower bound coefficient of 0.5 (i.e. a friction angle of 26.5°) was selected for $\theta_{int}$. Analysis with a friction coefficient increased to 0.6 led to a variation of the computed peak pulling force and bending moment (for $H/D = 3$) by only 1.5% and 3.7%, respectively.

The solver technique used was the direct full-Newton method with geometric nonlinearity continuously updated during the analysis. The first loading step features the application of geostatic stresses to establish the initial stress and strain states of the system owing to soil self-weight. Automatic incrementation of the displacement load was selected to achieve enhanced performance with regard to computational cost and numerical convergence.

## 3.4. Boundary conditions

Owing to the symmetry of the problem, only one half of the pipe and the surrounding soil is modelled together with appropriate boundary conditions: symmetry across the mid-plane of the structure, no

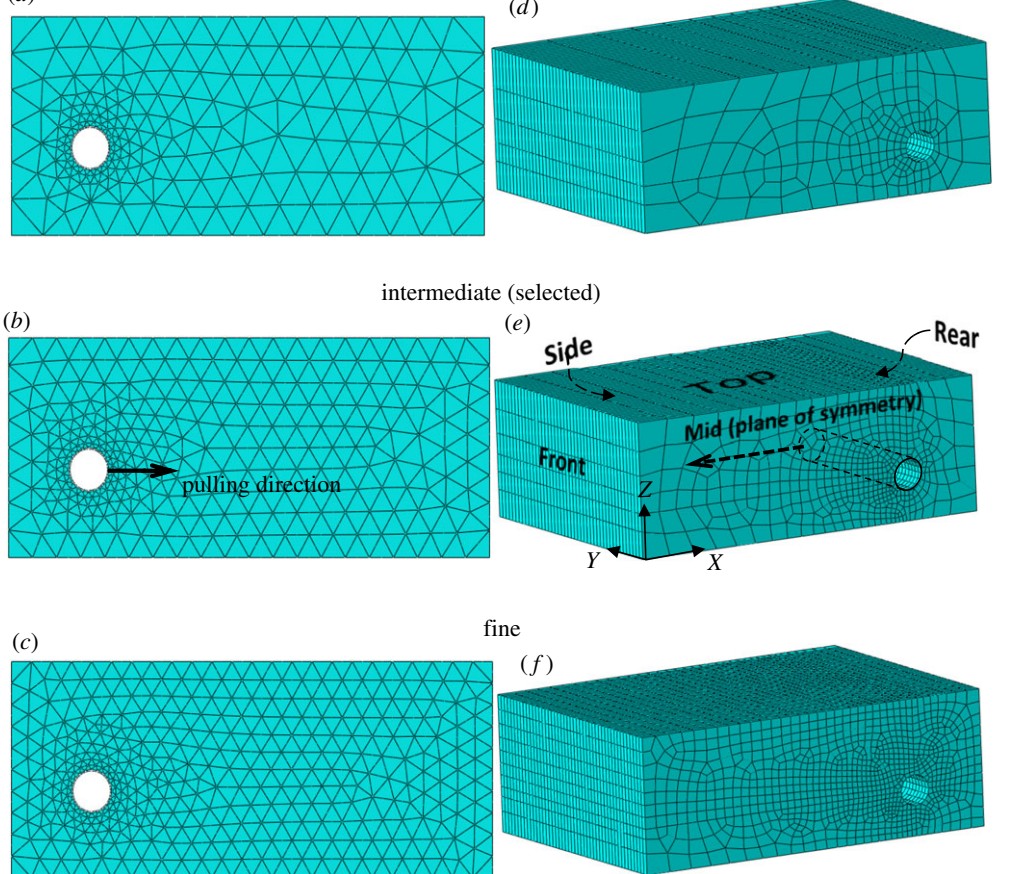

**Figure 4.** Mesh refinements analysed (*a*–*c*: 2D mesh and *d*–*f*: 3D mesh): (*a*) 360 elements, 782 nodes, (*b*) 548 elements, 1172 nodes, (*c*) 740 elements, 1568 nodes, (*d*) 7696 elements, 9154 nodes, (*e*) 19 620 elements, 21 934 nodes and (*f*) 38 910 elements, 42 364 nodes.

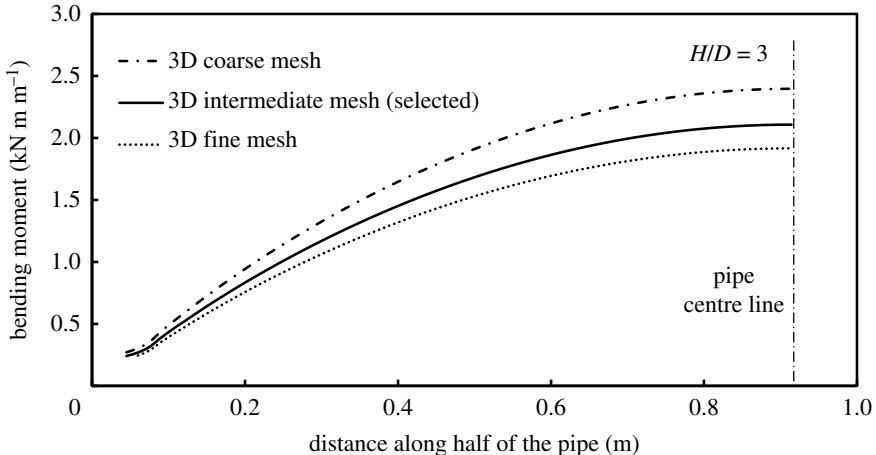

**Figure 5.** Mesh sensitivity effects on bending moment distribution along the pipe (3D_Linear Elements).

translations in the *x*-direction for both the front and rear faces of the soil block, no translations in the *y*-direction for the side face of the soil, and no translational movements across the bottom boundary of the soil block (figure 4*b*). A total prescribed horizontal displacement of one full pipe diameter was then imposed on the pipe end, and it was allowed to move freely in all other directions. The loading rate during the tests was relatively low (5 mm min$^{-1}$) [24], suggesting that inertia effects can be neglected. For this reason, quasi-static loading conditions were applied using the ABAQUS/CAE implicit solver. The calculated pulling load on the pipe was extracted from the reaction forces on the pipe end nodes where the prescribed displacement was applied.

## 3.5. Defining pipe displacement at peak pulling load

Because the current FEA does not account for strain softening of the soil (loss of strength with increasing strain), and the effect of reduced dilation angle at high strain levels is neglected, for most cases that were simulated, the identification of the peak strength from the load–displacement curve is rather cumbersome. This peak pulling force is an output of major importance either for pipe design or for comparison against experimental results. Therefore, a consistent and robust approach is needed to identify this peak force from the calculated results. In this section, various techniques for estimating peak pulling force are discussed.

One first approach that deals with this issue is through the identification of the pipe displacement at the peak pulling load, so that this displacement can be used to select a characteristic force from the calculated load–displacement curve (one that corresponds to the peak pulling force observed in experiments if the approach is successful). However, in a 3D problem where movements at the ends of the pipe are greater than at 'mid-span' (the point midway between the ends being pulled) as a result of bending deformations, there is more than one deformation value, so end displacement at peak (denoted $Y_{u,e}$), and mid-span displacement at peak (denoted $Y_{u,m}$) need to be examined separately if a characteristic deformation is to be identified.

ASCE [20] reports a procedure where lateral, rigid, pipe displacement at peak force is given by the following expression:

$$Y_u = f_y \left( H + \frac{D}{2} \right),$$ (3.5)

with the dimensionless parameter $f_y$ taking values between 0.02 and 0.03 for dense sand. Equation (3.5) is for plane strain loading, where the pipe reacts as a rigid body [22], or where it is very stiff compared to the soil and therefore behaves almost as a rigid body [17]. Hence, it can be concluded that for rigid pipes, $Y_{u,m}$ is equal to $Y_{u,e}$. The authors [24] have reported test results for a flexible pipe where end displacements were substantially higher than mid-span displacements. To use the ASCE [20] formula in situations where the pipe exhibits flexural behaviour, the calculated displacement would be more applicable to the mid-span of the pipe $Y_{u,m}$ (where plane strain conditions apply). Figure 6 compares the empirical formula according to ASCE [20] for dense sand (using the average value of $f_y = 0.025$) and experimental data found in the literature. Also included in the figure the displacement values corresponding to peak load measured at the mid-span (resembling plane strain behaviour of rigid bodies) for the pipe tested by Almahakeri *et al.* [24] which are modelled in the present study. The ASCE [20] formula overestimates the measured $Y_{u,m}$ values by Almahakeri *et al.* [24] by 11.1%, 28.5% and 16.7% for embedment ratios of 3, 5 and 7, respectively. To maintain consistency between the load–displacement curves used in the current research study and previous research works, both experimental and numerical data are plotted at pipe ends, and a modified $f_y$ value was fitted to account for longitudinal bending of flexible pipes. Figure 7 shows the peak displacement ratio ($Y_{u,e}/D$) at the pipe end versus embedment ratio for this study using a fitted value for $f_y = 0.065$. As can be observed, the fitted $f_y$ value yields a prediction that agrees very well with the measured displacements at peak load at the pipe end for all three examined embedment ratios. Further research is needed to develop a general formula that accounts for the influence of other potential factors that characterize 3D on $f_y$. Such factors may include $L/D$ ratio, $D/t$ ratio and the elastic modulus of the pipe. An illustration of how peak pulling force $P_u$ values corresponding to $Y_{u,e}$ are extracted from three different numerical analyses appears in a subsequent figure (figure 9) for an embedment ratio of 3. Additionally, in this figure and all figures containing the experimental load–displacement curves reported by Almahakeri *et al.* [24], the pipe displacement has been shifted to the left to account for the compliance of the pulling system (slack). The portion of the curve that has been rectified by this adjustment (the initial concave up section) is shown in grey (see inset in figure 9). A detailed discussion of this figure is presented in the 'Results and Discussion' section of this paper.

## 3.6. Relevance of 2D analysis for a 3D problem

When a section of pipeline is subjected to forces associated with lateral soil movements, design calculations are needed to estimate the bending stresses that result and the impact of those stresses on pipe performance. Given the difficulty of undertaking 3D analysis of the soil–pipe interaction, a computationally efficient approach is to simulate the behaviour as a plane strain problem to estimate

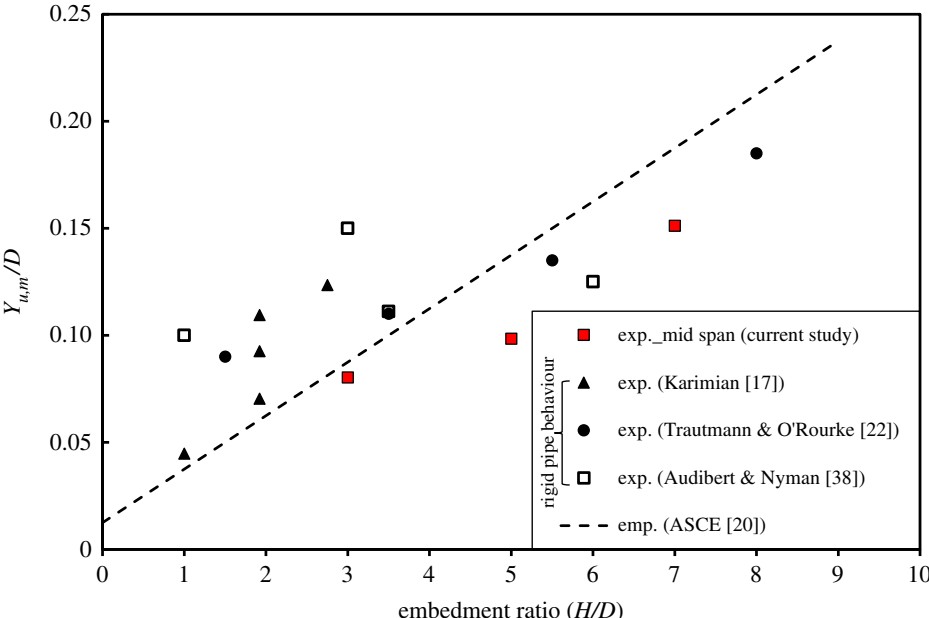

**Figure 6.** $Y_{u,m}/D$ versus embedment ratio ($Y_{u,m} = Y_{u,e} = Y_u$ for rigid pipes). exp., experimental value; emp., empirical expression ($f_y$ value is based on plane strain conditions).

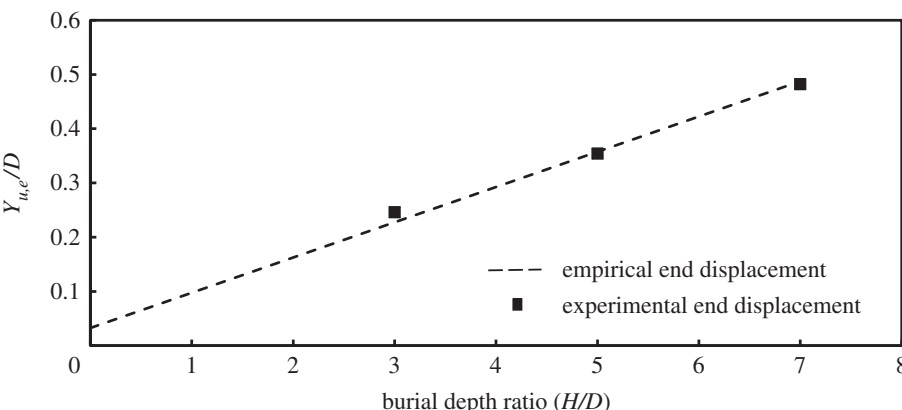

**Figure 7.** $Y_{u,e}/D$ versus embedment ratio (current study) (the empirical expression is based on a modified $f_y$ value to account for the flexibility of the pipe).

the lateral forces and bending moments. This approximation involves considering the lateral soil movements as being uniform along the pipeline, which means that the effect of longitudinal bending deflections along the pipe is neglected. The current study seeks to develop a simplified 3D analysis method, but also includes more conventional 2D calculations for comparison purposes (to assess the relative ability of these two different approaches to capture the peak pulling forces and pressure distribution around and along the pipe). In order to facilitate the comparisons between these two types of analysis, pulling load ($P$) is presented in this article in units of load/unit length (the units resulting from 2D calculations), unless otherwise stated, despite the fact that load is not generally uniform in laterally loaded pipes subjected to bending (the 3D results, therefore, represent the average magnitude of load per unit length along the whole of the pipe being modelled).

Figure 8$a$–$c$ show the contact pressure distribution around the pipe circumference for the 2D analysis compared with the pressure distribution at the same load level obtained from the 3D analysis at the pipe mid-span. The three figures covering analyses for $H/D = 3$, 5 and 7 suggest that analyses using 3D models provide comparable calculations to the 2D models if the contact pressure is averaged around the pipe mid-span circumference. The slight variation in pressure distribution shapes between the 2D

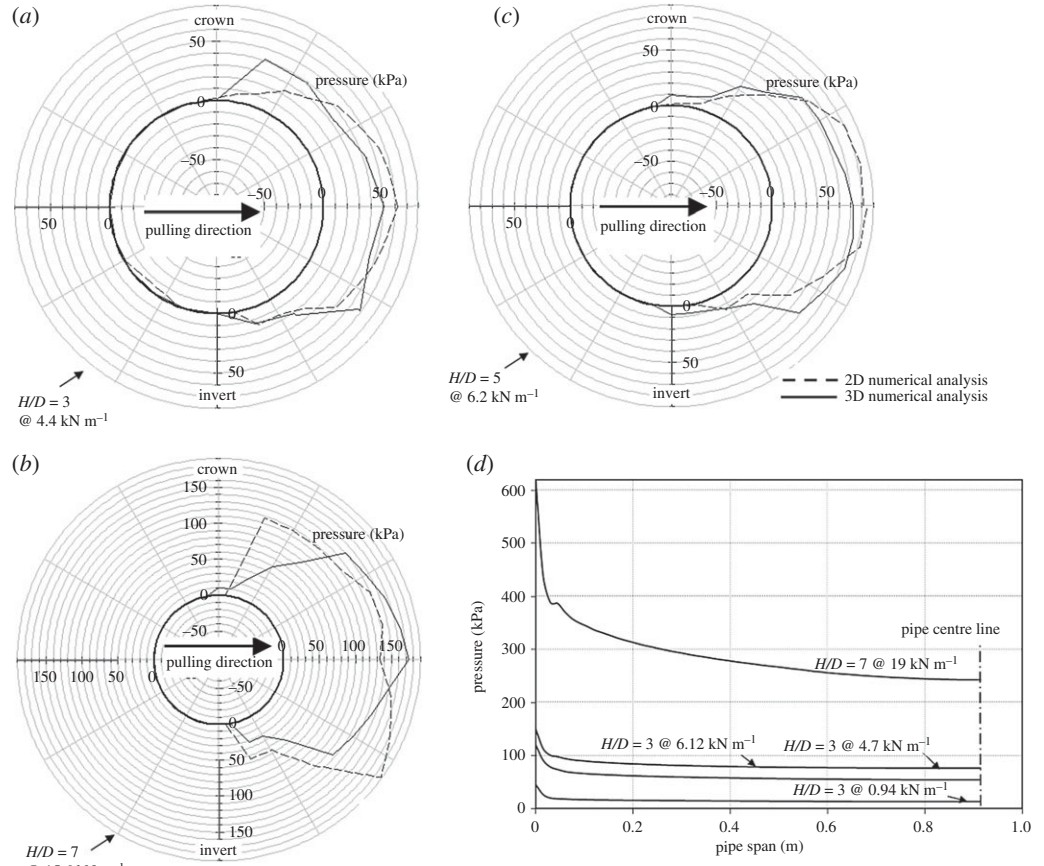

**Figure 8.** Lateral contact pressure distribution against pipe: ($a-c$) along the circumferential direction, ($d$) along the spring line direction.

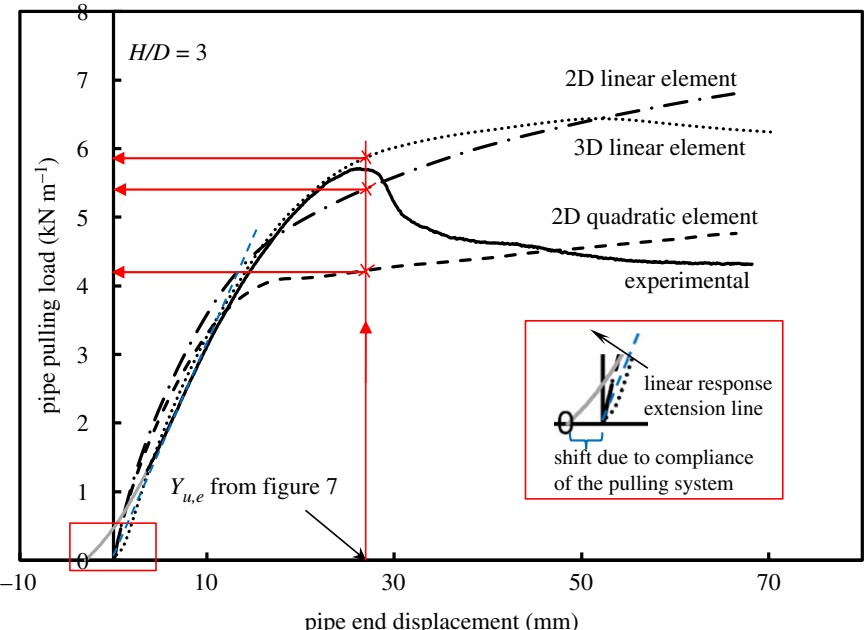

**Figure 9.** Measured and calculated values of pulling load versus pipe end displacement: measured (continuous curve) and calculated (dashed curves).

and 3D analyses can be attributed to the inherent differences between the two meshes used (element type, and the mesh refinement levels). On the other hand, examination of the variation of the longitudinal pressure distribution along the pipe spring line reveals that for relatively stiff pipes, the

pressure distribution varies with the change of confining pressure. At burial depth of $H/D = 3$, the pressure is fairly uniformly distributed along the pipe when pulling loads are still at relatively low values ($0.94\,\text{kN m}^{-1}$) as can be seen in figure 8$d$. However, the nonlinearity of pressure distribution increases for pulling loads between 4.7 and $6.12\,\text{kN m}^{-1}$. A more pronounced nonlinear pressure distribution is shown in the figure for $H/D = 7$ at a pulling load of $19\,\text{kN m}^{-1}$. These results suggest that a 2D analysis could still be used for the purpose of estimating loads imposed on the pipe from the perspective of soil capacity, but the nonlinearity of the pressure distribution along the pipe either with increase in pulling load or soil cover would emphasize the importance of conducting 3D analysis when an accurate prediction of the flexural behaviour of the pipe is required (as discussed further in the following sections).

# 4. Results and discussion

## 4.1. Numerical issues influencing 2D and 3D finite-element analysis results

There are a number of classical challenges associated with the use of FEA to obtain effective representations of shear failure mechanisms in soil. Factors influencing these include:

   (i) use of linear (lower order) and quadratic (higher order) displacement elements;
  (ii) the benefits of using lower order elements for modelling of master–slave interaction across boundaries;
 (iii) two-dimensional versus 3D analysis; and
 (iv) peak strength versus residual strength.

The manner in which each of these issues influences the calculations, and consequently, the development of the modelling approach used in this study, is discussed hereafter.

## 4.2. Influence of elements with linear versus quadratic displacement approximations

It is a well-known fact that higher order elements produce superior representations of shear failure for a variety of problems in soil mechanics [36]. Figure 9 includes:

— the measured load versus deflection for the experiment on the steel test pipe [24];
— a 2D calculation based on quadratic displacement (higher order) elements;
— a 2D calculation based on linear displacement (lower order) elements; and
— a 3D calculation based on linear displacement (lower order) elements.

Figure 9 also compares the performance of linear and quadratic elements for plane strain analysis of the buried pipe at residual strength of the soil, for an embedment ratio $H/D = 3$. These results are typical of what is calculated, whereby lower order elements overestimate the stiffness of the system and often fail to produce a clear peak load at higher displacements.

Because the present problem involves large deformations and the flow of soil around the pipe, the use of master–slave interaction is necessary. Even if contact pressures are uniform, higher order elements feature nodal forces that vary across the element (they are higher at mid-side nodes than at corners), and it is often necessary to use lower order elements to obtain stable solutions (as lower order elements produce uniform nodal forces across the element face) for this class of problem [32]. These difficulties are even more pronounced for 3D analyses (calculations using higher order elements in the 3D models did not converge from the early stages of the analysis). Therefore, all 3D analysis results reported herein are based on the use of lower order elements.

Figure 10 summarizes the comparison between all 2D calculations, to evaluate the performance of the two different types of finite elements (higher order and lower order) on the residual pulling force ($P_{\text{resid}}$) for the three examined burial depths, together with the results of 3D analysis using only the lower order elements (the issue of peak versus residual force is discussed in the next section). The figure shows that linear elements overestimate the soil resistance loads in all 2D numerical simulations compared to the experiments and relative to the loads obtained with quadratic elements by 34%, 27% and 22% for embedment ratios of 3, 5 and 7, respectively. The same figure also illustrates how 3D analysis produces higher values of $P_{\text{resid}}$ relative to the 2D analysis with linear elements for the three burial depths that were examined (by 7%, 14% and 9% for embedment ratios of 3, 5 and 7, respectively).

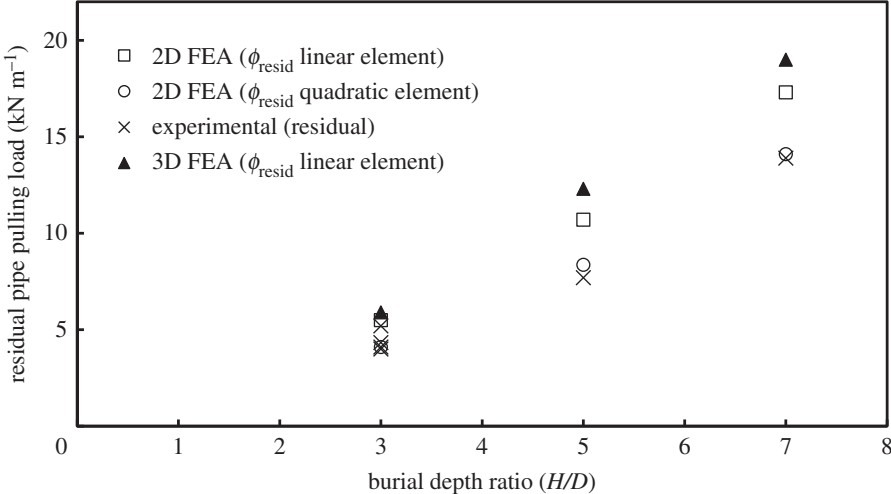

**Figure 10.** Residual pulling load using linear and quadratic elements.

This overestimation of the force observed in the 3D analysis compared to the 2D analysis could be attributed partly to the difference in mesh refinement level between the two analyses, though it might also be owing to other differences in the element formulations.

## 4.3. Estimation of peak versus residual pulling force

Another challenge associated with the analysis of the soil tested is associated with the dilation that occurs in dense sand during plastic shear deformation and the reductions in soil density and friction angle that result. This means that soil strength decreases as the soil undergoes increasing shear deformation. The conventional approach for solving such soil mechanics problem is to employ peak friction, $\phi_\mathrm{p}$, angle to estimate peak strength and residual angle of friction, $\phi_\mathrm{resid}$, to estimate residual strength. Provided the level of dilation in the soil undergoing shear deformation is reasonably uniform, calculations based on peak friction angle will not greatly exceed the peak measured strength. The use of uniform values of peak and residual friction angle is explored herein to identify to what extent the approach provides accurate predictions of the experimental response. However, the soil along the pipe exhibits complex 3D softening behaviour as strength varies along the pipe, with lower strength values being expected at the ends (because lateral movements are higher) and higher values being expected at mid-span (where lateral movements are smaller).

Figure 11 shows measured load versus displacement for $H/D = 3$, together with 2D calculations using higher order elements (quadratic displacement) based on both $\phi_\mathrm{resid}$ and $\phi_\mathrm{p}$. It is illustrated that the use of the peak friction angle, $\phi_\mathrm{p}$, provides a reasonable estimate of peak pulling force and that the residual friction angle, $\phi_\mathrm{resid}$, can be used to estimate the residual pulling force with sufficient precision. Figure 12 summarizes the performance of the 2D analysis based on higher order elements (with $\phi_\mathrm{p}$), along with calculations employing lower order elements (with $\phi_\mathrm{resid}$) for estimating peak pulling force for the three burial depths that were modelled. It is also shown how closely the 2D analyses can estimate peak soil load for both shallow and deeper burial conditions (1%, 5% and 7.8% differences relative to measured values for the embedment ratios of 3, 5 and 7, respectively). On the other hand, the 3D analysis showed similar behaviour but with higher differences (1.7%, 17.5% and 17.4% for the embedment ratios of 3, 5 and 7, respectively). It should be noted that the use of lower order 3D elements leads to load overestimation compared to second-order elements. However, the results shown in figure 12 for the 3D analysis are based on residual soil strength (i.e. $\phi_\mathrm{resid}$), which 'underestimates' the load compared to calculations based on $\phi_\mathrm{p}$. These effects compensate for each other, over the burial depth range considered here, and are discussed further in a subsequent section.

## 4.4. Load–displacement curve

Figures 13 and 14 show the measured load–displacement curves for the three burial depths and the numerical calculations using $\phi_\mathrm{resid}$ based on 2D and 3D FEAs, respectively. The figures illustrate how the peak pulling force of the soil $P_u$ increases as the embedment ratio increases. It can also be observed that

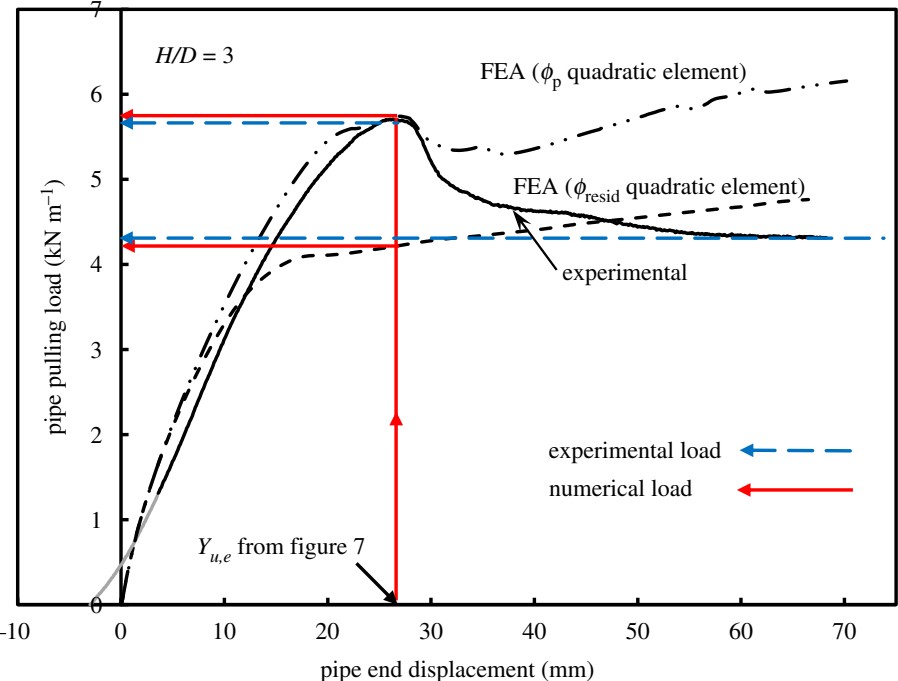

**Figure 11.** Effect of internal friction angle ($\phi$) on load–displacement curve: peak versus residual (2D quadratic elements).

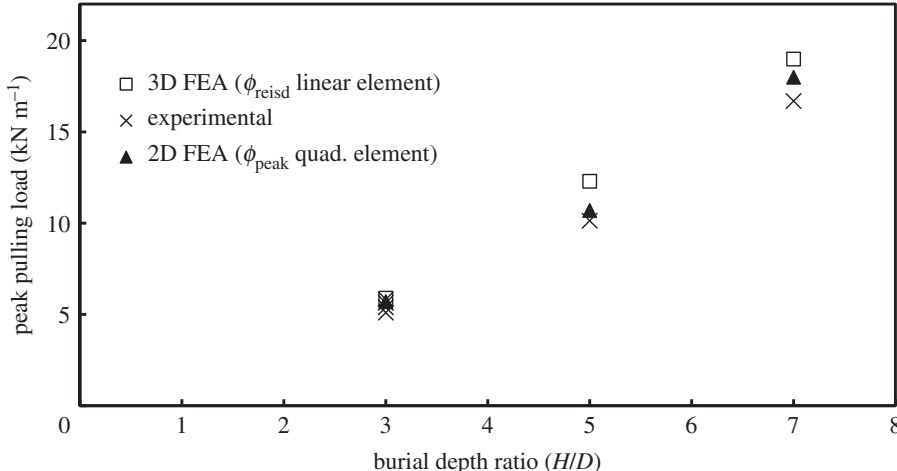

**Figure 12.** Peak pulling load versus burial depth ratio; analysis using 2D quadratic elements with $\phi_p$ and 3D linear elements with $\phi_{resid}$.

both the 2D and 3D analyses are close to the initial slope of the load–displacement curves for shallow cover ($H/D = 3$), while the predicted slope gets higher (i.e. stiffer response) as the burial depth increases. This higher stiffness response can be attributed partly to numerical limitations encountered when applying very low modulus of elasticity for the soil—near the surface—at greater burial depths ($H/D = 5$ and 7) as opposed to the theoretical modulus distribution calculated using the Janbu model (figure 3). Also, the soil modulus of elasticity used in the analyses is applied as an initial distribution and remains constant during the analysis, whereas in actual test conditions, soil modulus would be updated during the analysis. If such an approach was followed, as the soil was sheared (i.e. it weakens), the soil modulus would reduce and so would its stiffness (see [1] for further discussion).

## 4.5. Comparison to existing guidelines

This section treats the comparison of measured and calculated values (3D analysis) for $P_u$ reported in this study to calculated values according to the ASCE guidelines [20] based on the work of Trautmann &

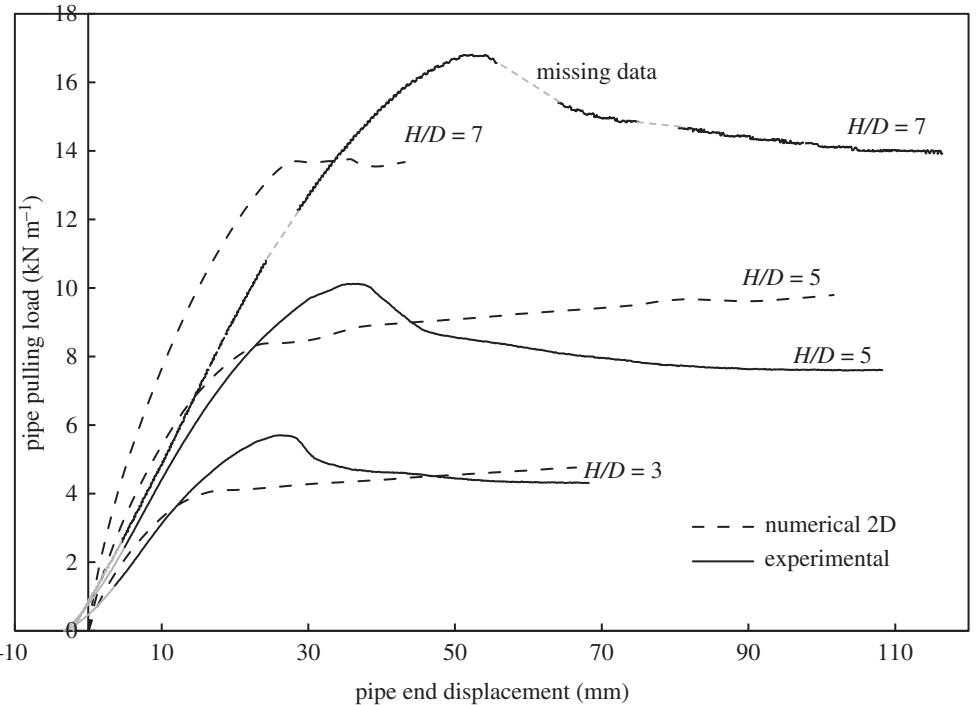

**Figure 13.** Load versus displacement curves at different burial depths; comparison of measurements to 2D calculations for $\phi_{\text{resid}}$.

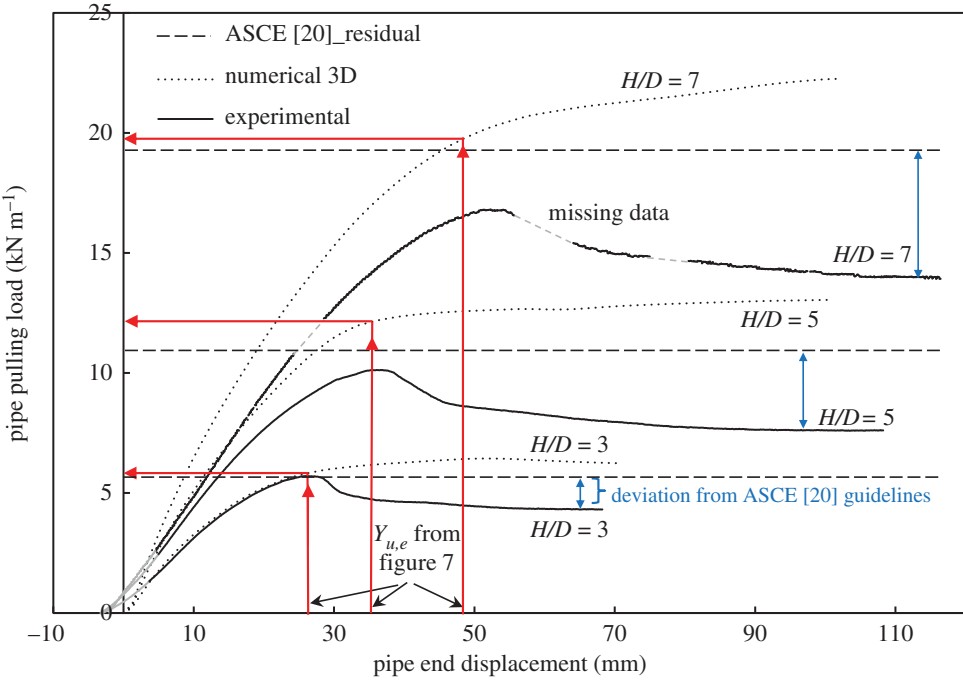

**Figure 14.** Load versus displacement curves at different burial depths; comparison of measurements to 3D calculations for ($\phi_{\text{resid}}$).

O'Rourke [22]. Even though the chart provided in those guidelines is based on the peak strength of the soil ($\phi_{\text{P}}$), the peak friction angle of the soil being studied herein ($\phi_{\text{p}} = 53°$) is outside the range provided in the ASCE [20] chart ($30°–45°$). For this reason, figure 14 is based on the residual strength ($\phi_{\text{resid}} = 45°$). In order to ensure direct comparison between the two methods, the ASCE guidelines residual strength estimations were compared to the residual strength of the soil as illustrated in figure 14. This comparison shows that residual lateral forces from the ASCE [20] approach exceed the experimental results by 25%, 53% and 42% for the embedment ratios of 3, 5 and 7, respectively, and are rather close to the predictions of the numerical calculations.

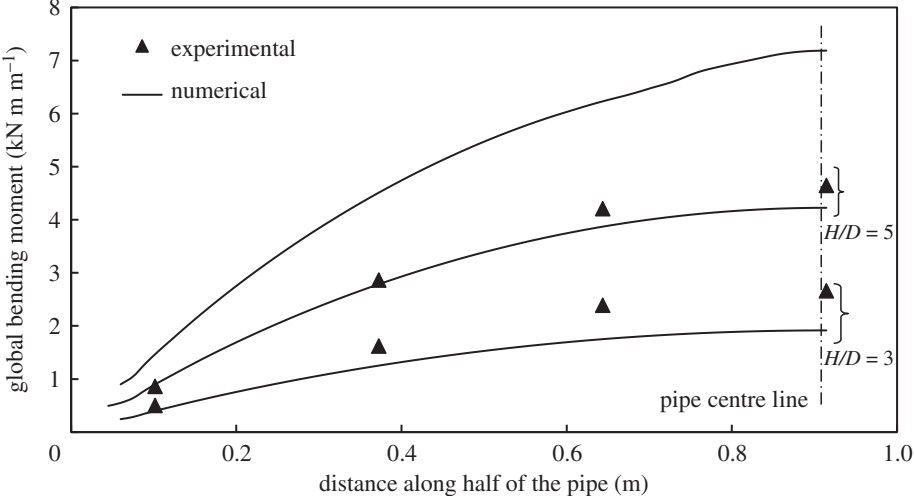

**Figure 15.** Pipe bending moment distribution at $Y_u$.

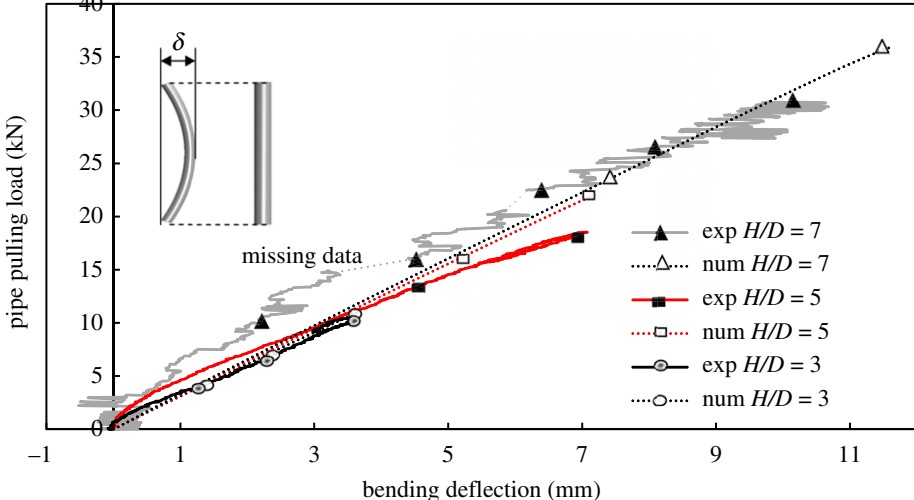

**Figure 16.** Progress of pipe bending deflection ($\delta$) at mid-span (exp: experimental data, num: numerical calculations).

## 4.6. Flexural behaviour of the pipe

In the previous sections, it was demonstrated that effective calculations of load versus displacement can be obtained using 3D analysis, where lower order elements are employed together with residual shear strength, $\phi_{resid}$. Such 3D analysis can be used to examine the flexural response of the pipe, quantified using curvature and bending moment distributions, the difference between the mid-pipe deflection and the movement at the pipe ends (classified here as the 'bending deflection', $\delta$), and the longitudinal strains at the extreme fibres (leading and trailing spring lines of the pipe). In each case, comparisons are made between calculated and measured values [24].

Figure 15 shows the measured and the calculated bending moment distribution along the pipe half-length at peak pulling load $P_u$, where the abscissas are expressed relative to the pipe end. The figure shows that the numerical analysis can predict reasonably well the bending moment distribution for embedment ratios $H/D$ of 3 and 5, with peak bending moment (at mid-span) underestimated by 20% and 8% for embedment ratios 3 and 5, respectively. Because the strain gage measurements experienced instabilities for the test performed at an embedment ratio of 7, no experimental data are available for comparison for this case [24].

Figure 16 shows both calculated and measured results for the load–bending deflection relation at pipe mid-span for the three burial depths that were examined. It is shown that the numerical model provides rather accurate estimates of the maximum deflection of the pipe with values overestimating the experimental ones by 1.7%, 4.3% and 10.3% for embedment ratios 3, 5 and 7, respectively.

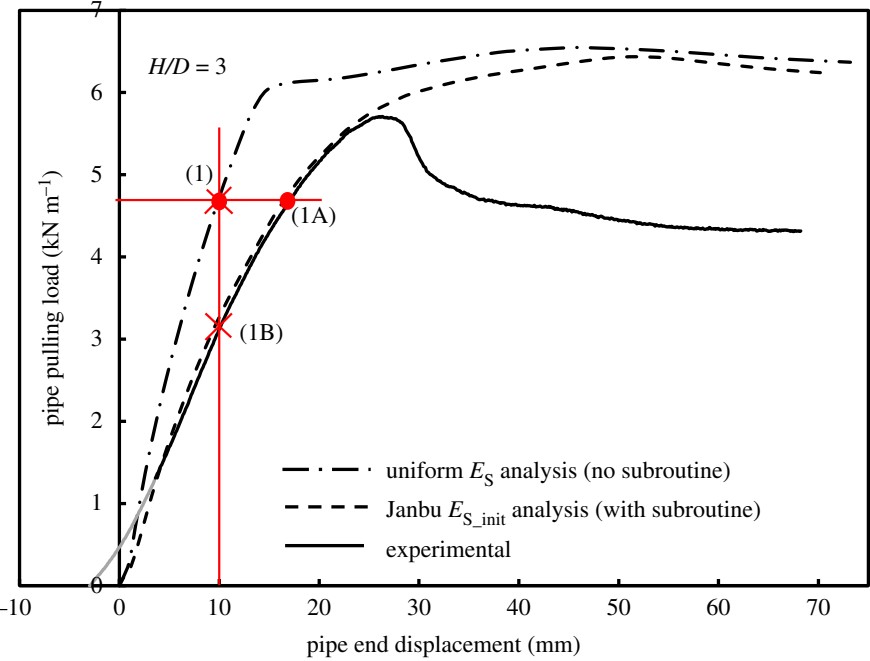

**Figure 17.** Effect of varying soil modulus on load–displacement curve.

However, the calculated bending deflection develops linearly as the ends of the pipe are pulled through the soil, while the test observations showed consistent deflection nonlinearity (curves that are concave down) for all three burial depths that were tested, i.e. there is less incremental load needed to produce the same incremental deflection of the pipe, even though the pipe material is believed to still be behaving elastically (based on the measured strains). This supports the idea that pipe bending behaviour, when confined in soil, is different from the flexural response 'in-air' [29].

## 4.7. Effect of soil modulus dependency on confining pressure

The effect of soil modulus that varies with depth based on the Janbu stress function (denoted 'Janbu initial' subsequently) on the flexural pipe response is illustrated in figure 17 using load–displacement curves and bending moment distributions for soil zones where soil modulus is uniform with depth according to equation (3.4) ($E_s = 803$ kPa used for $H/D = 3$), and the analyses reported earlier based on soil with the 'Janbu initial' modulus. The experimental results are also included. As discussed earlier, the numerical model in the current study focuses on the stress–strain behaviour up to the failure of the soil mass. Therefore, comparisons between calculated and measured results focus on the elastic loading range for the soil. For a given load value, for instance, 4.6 kN m$^{-1}$ (points '1' and '1A' in figure 17), the maximum bending moment at pipe mid-span using the 'Janbu initial' analysis (curve '1A' in figure 18) is 11% higher than the calculation based on uniform $E_s$ (curve '1'), but fitting very well to the experimental results. Furthermore, if the comparison is performed through consideration of a given pipe end displacement, for instance, 10 mm (points '1' and '1B' figure 17), the analysis with uniform $E_s$ overestimates the measured maximum bending moment at mid-span of the pipe by 20%, while the 'Janbu initial' analysis (curve '1B' in figure 18) provides an almost identical result to the experimental one. This finding suggests that there is a measurable difference between cases where soil modulus variation is taken into account versus analyses based on constant modulus considerations [1]. Analyses have been repeated for the $H/D = 5$ and 7 and the effect of soil modulus variation was very similar to the case of $H/D = 3$ presented here. This examination of pipe bending at loads before the peak suggests that the proposed numerical model provides calculations of pipe bending that are closer to the experimental results than those made at the peak (i.e. at the displacement of $Y_u$).

## 4.8. Additional experimental validation of the model with subroutine

The finite-element model—with subroutine—was also used to simulate the experiments of Karimian [17] for validation purposes. Those tests featured rigid steel buried pipes displaced laterally in dense uniform

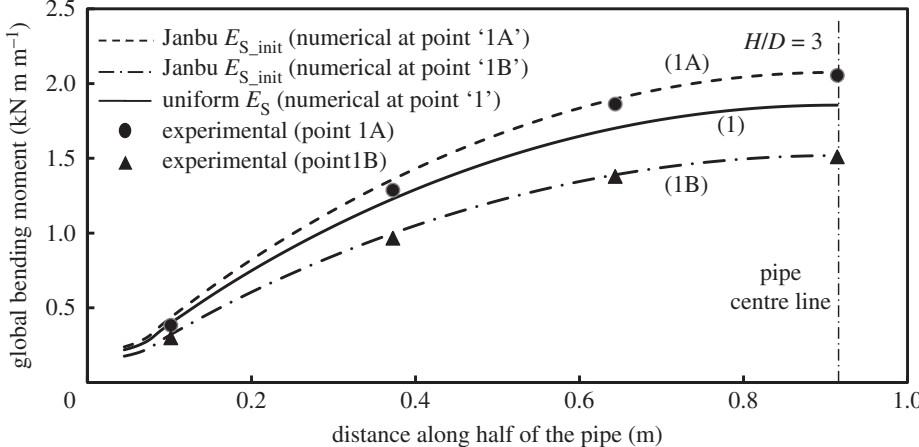

**Figure 18.** Effect of varying soil modulus on pipe bending moment distribution: at pipe loading of 4.6 kN m$^{-1}$ (curve '1A'), and at pipe end displacement of 10 mm (curve '1B').

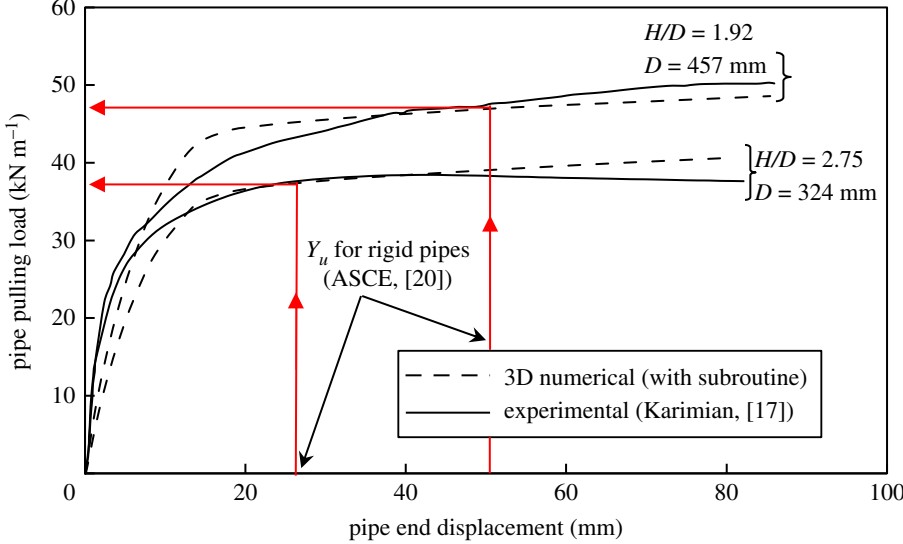

**Figure 19.** Validation of the numerical model on load–displacement curve.

Fraser River sand [17]. Figure 19 shows load–displacement curves of the 3D numerical analyses versus the experimental data for two of the tests featuring different pipe diameters (457 mm and 324 mm), each tested at a different embedment ratio ($H/D = 1.92$ and 2.75, respectively). In order to calculate $E_{S\_init}$ (equation (3.4)), the Janbu parameters needed to be calculated from the triaxial test data as reported by Karimian [17]. The $n$ and $K$ values were found to be 0.81 and 874, respectively. Other soil properties such as $\gamma = 16.3$ kN m$^{-3}$, $\phi_{resid} = 33°$ and $\psi = 15°$ are taken from the same study [17]. It is worth being reminded that the residual strength angle of friction of the soil ($\phi_{resid}$) has been used in the simulation instead of the peak value. Moreover, the numerical issue associated with the assignment of a minimum, non-zero, value of the soil modulus (to avoid numerical instability) was investigated in this validation work as well. The minimum modulus value to run an analysis was found to be 200 kPa and was applied to the top 10 mm of the soil cover.

Using the ASCE guidelines [20] (with $f_y = 0.03$ for rigid pipes), the pipe displacement at peak pulling load ($Y_u$) was calculated to be 23.8 and 44.6 mm for $H/D = 1.92$ and 2.75, respectively. Over the pre-peak load region of soil response, figure 19 shows that the numerical model underestimates the initial soil stiffness (occurring up to pipe displacement of 7% and 10% of pipe diameter for $H/D = 1.92$ and 2.75, respectively), while it overestimates the soil stiffness during the second half of that region. Peak loads were 0.61% (below) and 0.64% (above) the experimental values for $H/D = 1.92$ and 2.75, respectively. The numerical analyses reported by Karimian [17] produced similar results but with much stiffer initial soil behaviour when compared with the current study that adopts the 'Janbu initial' subroutine.

Such overestimated calculations reported by Karimian [17] for the loads lead consequently to overestimation of bending moment along the pipe as discussed in the previous section (curves '1' and '1B' in figure 18). Additional validation work of the current numerical model for flexible fibre reinforced polymer pipes has been discussed in detail by Almahakeri *et al.* [37].

## 5. Conclusion

The current study presents a 3D numerical modelling approach that is relatively simple to implement for 3D pipe–soil interaction problems. The numerical model also provides flexural bending stresses and strains along the pipe, together with the history of bending deflections up to the peak lateral force. A series of 2D and 3D FEAs have been conducted to simulate experiments where thin-walled steel pipes buried in dense sand were pulled laterally through the soil. The analyses were performed using ABAQUS/STANDARD, and employed large strain theory. The non-associated Mohr–Coulomb constitutive model was used for the sand with stress-dependent values of initial soil modulus (i.e. elastic soil modulus that increases with depth according to the Janbu stress function). An elastoplastic model was used for the steel pipe. While the numerical model proposed in the current study lacks the ability to capture the post-peak response of the soil since strain softening is neglected, the 3D model does provide good agreement with experimental results in estimating the soil response, load–displacement curves and the flexural behaviour of the pipe over the elastic loading range of the soil. Comparisons of the 2D and 3D calculations to the experimental observations lead to the following conclusions:

  (i) the 2D simulations using higher order elements can effectively provide the peak mobilized load (within 1%, 5% and 7% of the measured values for the embedment ratios of 3, 5 and 7, respectively) at the onset of plastic deformation of the soil;

 (ii) because the 3D analysis requires the use of lower order (linear displacement) elements, it overestimates the peak mobilized load (by an average of 43%). However, the model can effectively provide the progress of both the load–strain and load–deflection curves for the steel pipe, especially for embedment ratios $H/D = 3$ and 5. The agreement with experimental data measured over this embedment ratio range in the pre-peak pipe loading range would be of great interest for pipeline designers because most energy pipelines are buried at depths that range from one to four times the pipe diameter;

(iii) both 2D and 3D models are able to capture the effect of burial depth in terms of estimating pulling forces, and the 3D model provides the bending deflection of the pipe;

(iv) adopting a stress-dependent soil modulus improves significantly the numerical analysis calculations. For burial depth ratio $H/D = 3$, the numerical model produces load–displacement curves within 0.3% of the measured behaviour, and within 0.4% of the bending moment at $\frac{1}{4}$ diameter of pipe displacement; and

 (v) it is found that effective estimates of peak pulling force and bending moments along the pipeline can be obtained using 3D analysis based on residual shear strength. Such an approach is more straightforward and useful for buried pipe calculations compared to techniques that involve empirical adjustments of soil modulus or other calibrations.

Data accessibility. The datasets supporting this article have been uploaded as part of the electronic supplementary material.

Authors' contributions. I.D.M. and A.F. conceived the original idea for this study, provided feedback on the modelling, results analysis and manuscript draft. M.A. prepared and conducted the numerical simulations, analysed results and drafted the manuscript.

Competing interests. We declare we have no competing interests.

Funding. The work was funded by the Natural Sciences and Engineering Research Council of Canada, through Discovery Grants to A.F. and I.D.M.

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
