## [Reviewer comments · Royal Society Open Science]

Review History

RSOS-181550.R0 (Original submission)

Review form: Reviewer 1

Is the manuscript scientifically sound in its present form?

No

Are the interpretations and conclusions justified by the results?

No

Is the language acceptable?

No

Is it clear how to access all supporting data?

No

Do you have any ethical concerns with this paper?

No

Have you any concerns about statistical analyses in this paper?

I do not feel qualified to assess the statistics

Recommendation?

Major revision is needed (please make suggestions in comments)

Comments to the Author(s)

This paper provides numerical results from the analysis of pipes pulled horizontally through sand. The flow in the manuscript is weak in places and important information required in a paper focusing on modeling is missing. There is a disproportionate emphasis at the beginning on the experimental programme that had been carried out previously and published elsewhere. Given the large end displacements, up to 30 mm, progressive failure would be expected, which cannot be properly accommodated by the constitutive model. Some specifics include:

1. The English must be improved. In some places it is awkward, there are spelling mistakes and inappropriate words or grammatical tense are used. Take for example, the third sentence in the introduction. Soil instability may be considered a soil failure but soil creep is a mechanism. Also referring to 5 lines down from the top of page 2 the word focused should probably be replaced by focusing. On page 2, line 30: analyses is spelt incorrectly.
2. Figure 9 appears to be referred to before Figure 8.
3. It is clear from the introduction that the paper focuses on analysis, yet section 2 provides a disproportionate amount of information on the "Testing Program". From a numerical analysis point of view, it would have been sufficient to show the basic layout of the test being analyzed (not the facility) and the properties of the soil, including a figure of a typical stress-strain response of the sand. Had the sand been dense, it would probably have undergone post-peak softening that would have included shear banding. For such a case a constant dilation angle would have been inappropriate and local progressive failure would have taken place before the peak load would have been reached that reflects a macro-scale measure. The reviewer is aware that only the behavior up to peak was of interest.
4. The reviewer appreciates that a commercial code was used and that low-order elements are favoured for problems involving contact and large displacements. Were the stress and strain rate measures objective? Also, the 3-D 8-node element is known to have problems if bending modes are present, say adjacent to the pipe. Was selective (not reduced) integration used to improve the performance of the 8-node element?
5. Page 6, line 58: What does the quadrilateral element response have to do with the triangular element response? In section 4.2 it was made clear that the 2-D, 6-node triangular element was used. There are ways of predicting softer responses with 3-node triangular elements.
6. Page 7, Line 39: Should likely be referring to Figure 12 not 11.
7. Section 5.5: Just because ASCE charts go up to a friction angle of 45, is not a good reason to use the residual strength for carrying out the analysis.

Review form: Reviewer 2**Is the manuscript scientifically sound in its present form?**

Yes

Are the interpretations and conclusions justified by the results?

Yes

Is the language acceptable?

Yes

Is it clear how to access all supporting data?

Yes

Do you have any ethical concerns with this paper?

No

Have you any concerns about statistical analyses in this paper?

No

Recommendation?

Accept as is

Comments to the Author(s)

This paper presents experimental data and numerical simulations of tests investigating the behavior of buried steel pipelines subjected to lateral earth movement. The presented experimental data are useful for understanding pipeline behavior and for future numerical model calibration by other researchers. The presented numerical modeling, including the Janbu expression for the elastic modulus distribution with depth, is based on rather simple, standard procedures. The conclusions derived from the comparisons of the experimental measurements and the numerical results seem to be reasonable. Therefore, this reviewer recommends publication of the article.

Review form: Reviewer 3

Is the manuscript scientifically sound in its present form?

No

Are the interpretations and conclusions justified by the results?

Yes

Is the language acceptable?

No

Is it clear how to access all supporting data?

Yes

Do you have any ethical concerns with this paper?

No

Have you any concerns about statistical analyses in this paper?

Yes

Recommendation?

Major revision is needed (please make suggestions in comments)

Comments to the Author(s)

The paper needs substantial improvement in terms of writing, analysis and presentation before publishing. Some points to consider are as below. Subroutine incorporated modelling shows improvement in prediction of pipe response. But, the paper needs improvement by highlighting the its main contribution and significance.

- Please highlight the significance of research work (such as need of incorporating stress dependent E value)
- Line 46 p.1; it would be appropriate to use third person language in the manuscript
- Line 35 p.2: Please add more description of the tests; such as soil condition, soil type, etc
- Line 18 page4; cohesion of 1kpa can be excessive for dry sand. Please verify the selection
- Line 21 page4; please provide reference for dilation angle use
- Fig.5; large mesh sensitivity can be observed. Please define the mesh size in fine, medium and coarse mesh. Also, provide number of elements in each approach
- Section 4.5; please compare the model prediction of load-displacement curve with the experiments
- Please also compare peak load/(soil unit weight x height x diameter x length) as the guidelines are based on this scale
- Results and discussion section has not been written well. It needs improvement in terms of presentation and rigorous analysis.
- Would pipe pulling load same as pipe lateral load?
- It is clear from literature that plane strain and structured elements are the most suitable element type for this problem (lateral pipeline analysis). I am not sure why authors want to present this parametric study on element types? It appears not significant to present in the paper
- Why there is a substantial difference between tests and 3-D numerical model predictions in Fig 14
- Even with Janbu's incorporation, there is substantial difference in peak load and mobilization in Fig. 17. Please discuss
- It would be useful to propose analytical model for bending moment based on the numerical predictions using Janbu's stress dependent formulation

Decision letter (RSOS-181550.R0)

07-Mar-2019

Dear Dr Almahakeri,

The editors assigned to your paper ("NUMERICAL TECHNIQUES FOR DESIGN CALCULATIONS OF LONGITUDINAL BENDING IN BURIED STEEL PIPES SUBJECTED TO LATERAL EARTH MOVEMENTS") have now received comments from reviewers. We would like you to revise your paper in accordance with the referee and Associate Editor suggestions which can be found below (not including confidential reports to the Editor). Please note this decision does not guarantee eventual acceptance.

Please submit a copy of your revised paper before 30-Mar-2019. Please note that the revision deadline will expire at 00.00am on this date. If we do not hear from you within this time then it will be assumed that the paper has been withdrawn. In exceptional circumstances, extensions may be possible if agreed with the Editorial Office in advance. We do not allow multiple rounds of revision so we urge you to make every effort to fully address all of the comments at this stage. If deemed necessary by the Editors, your manuscript will be sent back to one or more of the

original reviewers for assessment. If the original reviewers are not available, we may invite new reviewers.

- Data accessibility

If you wish to submit your supporting data or code to Dryad (<http://datadryad.org/>), or modify your current submission to dryad, please use the following link:
<http://datadryad.org/submit?journalID=RSOS&manu=RSOS-181550>

- Competing interests

- Authors' contributions

- Acknowledgements

- Funding statement

Kind regards,

Andrew Dunn

on behalf of Prof R. Kerry Rowe (Subject Editor)

Associate Editor's comments:

Please accept our apologies for the delay in completing review of your manuscript: we struggled to recruit suitable referees as rapidly as we (and, no doubt, you) had hoped. Nevertheless, we have now secured the commentary of three reviewers. As you will see, there are a number of matters that need to be addressed before the manuscript may be considered further. Please incorporate the changes requested by the reviewers and, if you do not do so, provide full rebuttals for any absences -- if you are unable to satisfy the reviewers that the paper is ready for acceptance post-revision, we will not be able to consider it further for publication. Best of luck and we'll look forward to receiving your revision shortly.

Comments to Author:

Reviewers' Comments to Author:

Reviewer: 1

Comments to the Author(s)

This paper provides numerical results from the analysis of pipes pulled horizontally through sand. The flow in the manuscript is weak in places and important information required in a paper focusing on modeling is missing. There is a disproportionate emphasis at the beginning on the experimental programme that had been carried out previously and published elsewhere. Given the large end displacements, up to 30 mm, progressive failure would be expected, which cannot be properly accommodated by the constitutive model. Some specifics include:

1. The English must be improved. In some places it is awkward, there are spelling mistakes and inappropriate words or grammatical tense are used. Take for example, the third sentence in the

introduction. Soil instability may be considered a soil failure but soil creep is a mechanism. Also referring to 5 lines down from the top of page 2 the word focused should probably be replaced by focusing. On page 2, line 30: analyses is spelt incorrectly.

2. Figure 9 appears to be referred to before Figure 8.

3. It is clear from the introduction that the paper focuses on analysis, yet section 2 provides a disproportionate amount of information on the "Testing Program". From a numerical analysis point of view, it would have been sufficient to show the basic layout of the test being analyzed (not the facility) and the properties of the soil, including a figure of a typical stress-strain response of the sand. Had the sand been dense, it would probably have undergone post-peak softening that would have included shear banding. For such a case a constant dilation angle would have been inappropriate and local progressive failure would have taken place before the peak load would have been reached that reflects a macro-scale measure. The reviewer is aware that only the behavior up to peak was of interest.

4. The reviewer appreciates that a commercial code was used and that low-order elements are favoured for problems involving contact and large displacements. Were the stress and strain rate measures objective? Also, the 3-D 8-node element is known to have problems if bending modes are present, say adjacent to the pipe. Was selective (not reduced) integration used to improve the performance of the 8-node element?

5. Page 6, line 58: What does the quadrilateral element response have to do with the triangular element response? In section 4.2 it was made clear that the 2-D, 6-node triangular element was used. There are ways of predicting softer responses with 3-node triangular elements.

6. Page 7, Line 39: Should likely be referring to Figure 12 not 11.

7. Section 5.5: Just because ASCE charts go up to a friction angle of 45, is not a good reason to use the residual strength for carrying out the analysis.

Reviewer: 2

Comments to the Author(s)

This paper presents experimental data and numerical simulations of tests investigating the behavior of buried steel pipelines subjected to lateral earth movement. The presented experimental data are useful for understanding pipeline behavior and for future numerical model calibration by other researchers. The presented numerical modeling, including the Janbu expression for the elastic modulus distribution with depth, is based on rather simple, standard procedures. The conclusions derived from the comparisons of the experimental measurements and the numerical results seem to be reasonable. Therefore, this reviewer recommends publication of the article.

Reviewer: 3

Comments to the Author(s)

The paper needs substantial improvement in terms of writing, analysis and presentation before publishing. Some points to consider are as below. Subroutine incorporated modelling shows improvement in prediction of pipe response. But, the paper needs improvement by highlighting the its main contribution and significance.

- Please highlight the significance of research work (such as need of incorporating stress dependent E value)
- Line 46 p.1; it would be appropriate to use third person language in the manuscript
- Line 35 p.2: Please add more description of the tests; such as soil condition, soil type, etc
- Line 18 page4; cohesion of 1kpa can be excessive for dry sand. Please verify the selection
- Line 21 page4; please provide reference for dilation angle use

- Fig.5; large mesh sensitivity can be observed. Please define the mesh size in fine, medium and coarse mesh. Also, provide number of elements in each approach
- Section 4.5; please compare the model prediction of load-displacement curve with the experiments
- Please also compare peak load/(soil unit weight x height x diameter x length) as the guidelines are based on this scale
- Results and discussion section has not been written well. It needs improvement in terms of presentation and rigorous analysis.
- Would pipe pulling load same as pipe lateral load?
- It is clear from literature that plane strain and structured elements are the most suitable element type for this problem (lateral pipeline analysis). I am not sure why authors want to present this parametric study on element types? It appears not significant to present in the paper
- Why there is a substantial difference between tests and 3-D numerical model predictions in Fig 14
- Even with Janbu's incorporation, there is substantial difference in peak load and mobilization in Fig. 17. Please discuss
- It would be useful to propose analytical model for bending moment based on the numerical predictions using Janbu's stress dependent formulation

Author's Response to Decision Letter for (RSOS-181550.R0)

See Appendix A.

RSOS-181550.R1 (Revision)

Review form: Reviewer 3

Is the manuscript scientifically sound in its present form?

Yes

Are the interpretations and conclusions justified by the results?

Yes

Is the language acceptable?

No

Is it clear how to access all supporting data?

Not Applicable

Do you have any ethical concerns with this paper?

No

Have you any concerns about statistical analyses in this paper?

No

Recommendation?

Accept with minor revision (please list in comments)

Comments to the Author(s)

Authors have adequately addressed my first round comments and hence I suggest accepting. However, the structure and presentation of the manuscript can be improved.

Decision letter (RSOS-181550.R1)

10-May-2019

Dear Dr Almahakeri:

On behalf of the Editors, I am pleased to inform you that your Manuscript RSOS-181550.R1 entitled "NUMERICAL TECHNIQUES FOR DESIGN CALCULATIONS OF LONGITUDINAL BENDING IN BURIED STEEL PIPES SUBJECTED TO LATERAL EARTH MOVEMENTS" has been accepted for publication in Royal Society Open Science subject to minor revision in accordance with the referee suggestions. Please find the referees' comments at the end of this email.

The reviewers and Subject Editor have recommended publication, but also suggest some minor revisions to your manuscript. Therefore, I invite you to respond to the comments and revise your manuscript.

- Ethics statement

- Data accessibility

<http://datadryad.org/submit?journalID=RSOS&manu=RSOS-181550.R1>

- Competing interests

- Authors' contributions

- Acknowledgements

- Funding statement

Because the schedule for publication is very tight, it is a condition of publication that you submit the revised version of your manuscript before 19-May-2019. Please note that the revision deadline will expire at 00.00am on this date. If you do not think you will be able to meet this date please let me know immediately.

- 1) A text file of the manuscript (tex, txt, rtf, docx or doc), references, tables (including captions) and figure captions. Do not upload a PDF as your "Main Document".
- 2) A separate electronic file of each figure (EPS or print-quality PDF preferred (either format should be produced directly from original creation package), or original software format)

- 3) Included a 100 word media summary of your paper when requested at submission. Please ensure you have entered correct contact details (email, institution and telephone) in your user account
- 4) Included the raw data to support the claims made in your paper. You can either include your data as electronic supplementary material or upload to a repository and include the relevant doi within your manuscript
- 5) All supplementary materials accompanying an accepted article will be treated as in their final form. Note that the Royal Society will neither edit nor typeset supplementary material and it will be hosted as provided. Please ensure that the supplementary material includes the paper details where possible (authors, article title, journal name).

on behalf of Prof R. Kerry Rowe (Subject Editor)
openscience@royalsociety.org

Associate Editor Comments to Author:

The reviewer considers the paper to be scientifically sound and ready for publication, but you should take another look at the paper's presentation before resubmitting, as it appears additional work to improve the clarity of presentation might be possible. Nevertheless, congratulations on this outcome!

Reviewer comments to Author:

Reviewer: 3

Comments to the Author(s)

Authors have adequately addressed my first round comments and hence I suggest accepting. However, the structure and presentation of the manuscript can be improved.

Author's Response to Decision Letter for (RSOS-181550.R1)

See Appendix B.

Decision letter (RSOS-181550.R2)

31-May-2019

Dear Dr Almahakeri,

I am pleased to inform you that your manuscript entitled "NUMERICAL TECHNIQUES FOR DESIGN CALCULATIONS OF LONGITUDINAL BENDING IN BURIED STEEL PIPES SUBJECTED TO LATERAL EARTH MOVEMENTS" is now accepted for publication in Royal Society Open Science.

on behalf of Prof R. Kerry Rowe (Subject Editor)
openscience@royalsociety.org

Appendix A

1 Reviewer-1

1. Comments to the Author(s)

This paper provides numerical results from the analysis of pipes pulled horizontally through sand. The
flow in the manuscript is weak in places and important information required in a paper focusing on
modeling is missing. There is a disproportionate emphasis at the beginning on the experimental
programme that had been carried out previously and published elsewhere. Given the large end
displacements, up to 30 mm, progressive failure would be expected, which cannot be properly
accommodated by the constitutive model. Some specifics include:

The English must be improved. In some places it is awkward, there are spelling mistakes and
inappropriate words or grammatical tense are used. Take for example, the third sentence in the
introduction. **"Soil instability may be considered a soil failure but soil creep is a mechanism".**

*** Thank you for taking the time to review this manuscript. It is very much
appreciated!*

*Authors have reviewed the first three sentences of the introduction (shown
below) and could not find the incorrect sentence that is being referred to
in the comment. A comprehensive review of the manuscript has been conducted
and many improvements have been introduced to the manuscript in general.*

**"Oil and gas transmission lines can cross zones of soil instability and may need to be
designed to resist differential ground movements. Soil instabilities can result from natural
phenomena such as soil creep, slope failures, landslides, and earthquake-induced faults.
Other man-made activities can lead to lateral soil movements against pipes such as
excavation, tunneling, directional drilling, and pipe bursting"**

2. Also referring to 5 lines down from the top of page 2 the word focused should probably be replaced by
focusing.

*** "focused" has been changed to "focusing".*

3. On page 2, line 30: analyses is spelt incorrectly.

*** Authors used the plural form of the word not the singular in referral to
a class of 3D-FEA analyses that are often encountered when using lower-
order elements. Upon further review and considering the reviewer's feedback
on the readability of the sentence, the word has now changed to the
singular form "analysis".*

4. Figure 9 appears to be referred to before Figure 8.

*** The reason is that Figure 9 is the first figure where the method of
determining the pipe displacement at peak pulling load, Y_u , could be
illustrated. Also, it was necessary to present the concept of P_u - Y_u relation
before discussing Figure 8. To clear up any confusion this may cause to the
reader, the sentence has been modified as follows (Lines 266-267):*

“An illustration of how peak pulling force P_u values corresponding to $Y_{u,e}$ are extracted from three different numerical analyses appears in a subsequent figure (Figure 9), for embedment ratio of $H/D=3$.”

5. It is clear from the introduction that the paper focuses on analysis, yet section 2 provides a disproportionate amount of information on the “Testing Program”. From a numerical analysis point of view, it would have been sufficient to show the basic layout of the test being analyzed (not the facility) and the properties of the soil, including a figure of a typical stress-strain response of the sand.

*** The “Testing Facility” sub-section has been removed and the “Testing Program” section has been rewritten to provide a more concise description of the experimental work. Also, in addition to the experimental data (Load-Displacement curve) that depict the stress-strain behaviour of dense sand, a reference for additional details of the soil characterization tests including grain size analysis, and triaxial tests have been included in this section. It reads (Line 59):*

“Details of the soil characterization tests including grain size analysis, and triaxial tests are discussed by Almahakeri [25].”

6. Had the sand been dense, it would probably have undergone post-peak softening that would have included shear banding. For such a case a constant dilation angle would have been inappropriate and local progressive failure would have taken place before the peak load would have been reached that reflects a macro-scale measure. The reviewer is aware that only the behavior up to peak was of interest.

*** Authors do agree with the reviewer’s comment on dense sand behavior. As one of the main objectives of the current study (stated in Lines 40-41) is to develop a simple 3D modeling technique to estimate the load-deflection response of pipes moving laterally through the ground, based on independent parameters for the pipe and soil. This is achieved by describing a simple stress-dependent soil modulus -up to peak loading- using a built-in “utility” subroutine (by employing the Janbu model) and comparing it against different experimental results. Other, more sophisticated, full material user-defined, subroutines are more robust, but often not available in the public domain. The model presented in the current study, which has been verified against different experimental results can be of direct benefit to both researchers and industry practitioners alike. Authors can reflect this idea in the text if deemed necessary.*

7. The reviewer appreciates that a commercial code was used and that low-order elements are
favoured for problems involving contact and large displacements. Were the stress and strain
rate measures objective? Also, the 3-D 8-node element is known to have problems if
bending modes are present, say adjacent to the pipe. Was selective (not reduced) integration
used to improve the performance of the 8-node element?

*** ABAQUS software does incorporate the use of "selective Integration"
formulation for its 8-noded linear hexahedral elements (C3D8) formulation.*

8. Page 6, line 58: What does the quadrilateral element response have to do with the triangular
element response? In section 4.2 it was made clear that the 2-D, 6-node triangular element
was used. There are ways of predicting softer responses with 3-node triangular elements.

*** Upon review, the two sentences relevant to the quadrilateral elements
behavior and the reference to the work by Sloan (1983) does not apply to
the work presented in this manuscript since all 2D analyses have been
executed using the triangle elements (as indicated in section 4.2). While
attempts of using quadrilateral elements were made (discussed briefly in
the original submission-section 4.2, Lines 168-169), no detailed results
were reported. Hence, the following two sentences have been removed from
the manuscript.*

***"Sloan [36] explains that quadrilateral elements are known to be over stiff when used with
full integration, and that oscillations can arise when using reduced integration to counter
those problems. Sloan indicates that this influences both drained behavior (where there is
frictional behaviour and volume change) and undrained analysis (where friction angle is
zero and volume change should be zero"***

9. Page 7, Line 39: Should likely be referring to Figure 12 not 11.

*** Corrected.*

10. Section 5.5: Just because ASCE charts go up to a friction angle of 45, is not a good reason to
use the residual strength for carrying out the analysis.

*** The reason ASCE soil capacity limits were calculated based on the constant
volume friction angle ($\phi_{resid} = 45^\circ$) instead of peak friction angle ($\phi_p = 53^\circ$) is
to ensure direct comparison between the two methods. As no peak loads could
be extracted from ASCE guidelines for critical friction angle of 53° , authors*

compared the ASCE using ϕ_{resid} with the residual strength of the soil -not the
peak load- as shown in Figure 14.

A clarification has been introduced in the text as follows (Lines 382-384):

***“In order to ensure direct comparison between the two methods, the ASCE guidelines***
***residual strength estimations were compared to the residual strength of the soil as***
***illustrated in Figure 14.”***

***Thank you for your time and valuable feedback!

**Reviewer: 2**

Comments to the Author(s)

This paper presents experimental data and numerical simulations of tests investigating the
behavior of buried steel pipelines subjected to lateral earth movement. The presented
experimental data are useful for understanding pipeline behavior and for future numerical
model calibration by other researchers. The presented numerical modeling, including the Janbu
expression for the elastic modulus distribution with depth, is based on rather simple, standard
procedures. The conclusions derived from the comparisons of the experimental measurements
and the numerical results seem to be reasonable. Therefore, this reviewer recommends
publication of the article.

*** Thank you for taking the time to review this manuscript. It is very much*
*appreciated!*

**Reviewer: 3**

Comments to the Author(s)

The paper needs substantial improvement in terms of writing, analysis and presentation before
publishing. Some points to consider are as below. Subroutine incorporated modelling shows
improvement in prediction of pipe response. But, the paper needs improvement by
highlighting the its main contribution and significance.

*** Thank you for taking the time to review this manuscript. It is very much*
*appreciated!*

1. Please highlight the significance of research work (such as need of incorporating stress
dependent E value)

**** An introductory sentence has been added at the start of the literature*
*review section (Line 1,third paragraph) and another one at the end of the*
*section (Line 19)to highlight the significance and focus of the current study*
*on the use of stress-dependant model to the soil stress-strain relation. Both*
*sentences are shown below:*

***“In pipe-soil interaction problems, it is known that the soil stress-strain relation governs the***
***restraint loads imposed on the pipeline. Over the years, many models have been developed to***
***capture the influence of different parameters of the soil.***

.. Section on literature review goes here ..

***In the present study, simple 2D and 3D finite element models employing stress-dependent***
***stiffness of the soil using Janbu model subroutine are presented and discussed.”***

2. Line 46 p.1; it would be appropriate to use third person language in the manuscript

**** Three relative pronouns (“who”) were identified and removed from the*
*text.*

3. Line 35 p.2: Please add more description of the tests; such as soil condition, soil type, etc

**** In consideration for Reviewer #1 suggestion to keep some appropriation of*
*the size of the description of the experimental work (by reducing it) and to*
*address the current comment, authors have cited an additional reference for*
*readers to get more details about the testing program. Additionally, basic*
*soil conditions and parameters (dry Olivine dense sand, with friction angle,*
*dilation angle, density, and modulus of elasticity distribution) have been*
*reported in the manuscript. Authors can provide any other specific*
*information related to the soil related to the study.*

4. Line 18 page4; cohesion of 1kpa can be excessive for dry sand. Please verify the selection

*** Additional verification of the choice has been included in the text. New
sentence now reads (Lines 142-144):

**For cohesion, c , since the Mohr–Coulomb model implemented in ABAQUS requires nonzero**
**cohesion to accommodate the shape of flow potential close to the apex of the model, a**
**minimum value of 1 kPa was employed even though the test sand was dry. This artificial**
**cohesion was introduced to ensure the soil had a small but non-zero strength when in a state**
**of zero confining stress. This is a common requirement reported in many other studies**
**[3,30,31].**

5. Line 21 page4; please provide reference for dilation angle use

*** Reference has been added to text.

6. Fig.5; large mesh sensitivity can be observed. Please define the mesh size in fine, medium and
coarse mesh. Also, provide number of elements in each approach

*** Additional information has been added to indicate the element size (Line
189), and node seeding approach used in the finite element models [Line 183].
Also, additional figures (Figure 4a,b,c,d,e, and f) showing the different
meshes used for the analyses (2D &3D) along with the corresponding number of
nodes and elements have been added.

7. Section 4.5; please compare the model prediction of load-displacement curve with the
experiments

*** Authors reviewed section 4.5 and there was no mention about load-
displacement curves. The section primarily discusses the method used to
define pipe displacement at peak pulling load (Y_u). If the comment is
regarding Y_u , both Figure 6 and Figure 7 include comparison with the
experiments. For load-displacement curves, Figures 9, 11, 13, and 14 compare
the different model calculations to the experimental results.

8. Please also compare peak load/(soil unit weight x height x diameter x length) as the guidelines
are based on this scale

*** Authors reported peak loads (per unit length of the pipe) to maintain
consistency with the other load-displacement figures in the manuscript. A
clarification has been included in the text to indicate that load-
displacement curves in Figure 14(experimental and numerical) are compared to

corresponding values (i.e., pulling load/unit length) using the ASCE (1984)
guidelines. New text now reads (Lines 379-380):

**“Measured and calculated values for P_u reported in this study are now compared with the**
**corresponding values using ASCE guidelines [20] based on the work of Trautmann and**
**O’Rourke [22]”**

9. Results and discussion section has not been written well. It needs improvement in terms of
presentation and rigorous analysis.

*Authors assume that the revised manuscript has now sufficient analysis and*
*clarification.*

10. Would pipe pulling load same as pipe lateral load?

*Yes, it is the same load as pulling displacement was applied at the pipe end*
*nodes (where reaction forces were calculated). A clarification has been added*
*to the text. Lines (229-230) now read:*

**“The calculated pulling load on the pipe was extracted from the reaction forces of the pipe**
**end nodes where prescribed displacement was applied.”**

11. It is clear from literature that plane strain and structured elements are the most suitable
element type for this problem (lateral pipeline analysis). I am not sure why authors want to
present this parametric study on element types? It appears not significant to present in the
paper

**** Plane strain elements have been used in the current study as indicated in*
*section 5.2 (Line 166). The use of the automatic built-in mesh generation of*
*the software was employed as the structured mesh configuration encountered*
*numerical instability for the CPE6M elements. However, there was a negligible*
*difference between the two different mesh configurations over the completed*
*part of the analysis. The above comment has been addressed in the text. Lines*
*(185-187) now read:*

**“The automatic built-in mesh generation of the software was employed as structured mesh**
**configuration encountered numerical instability for the CPE6M elements (analysis aborted at**
**10.3% of completion. Comparison of the load-displacement curve between the two mesh**
**configurations over the completed part of the analysis showed almost identical behavior.”**

**12.** Why there is a substantial difference between tests and 3-D numerical model predictions in Fig

14

*The explanation was provided in the original manuscript (now lines 371-376)*

**13.** Even with Janbu's incorporation, there is substantial difference in peak load and mobilization in

Fig. 17. Please discuss

*Figure 17 is the comparison between the calculated load-displacement curve*
*using Janbu model and the one using uniform soil modulus. The Janbu model*
*curve (shown in Figure 17) was first introduced and discussed in an earlier*
*section. It was also compared to experimental data (Figure 9). It is worth*
*noting also that peak loading was determined using the Y_u discussed in*
*section 4.5. and hence, Janbu model showed very good agreement at the burial*
*test illustrated in Figure 17.*

**14.** It would be useful to propose analytical model for bending moment based on the numerical
predictions using Janbu's stress dependent formulation

*Thank you for the suggestion. This definitely is an excellent topic for*
*future work authors can explore in future work.*

****Thank you for your time and valuable feedback!*

Appendix B

June 25, 2019

Editor

Royal Open Society

Dear Editor:

RE: Numerical Techniques for Design Calculations of Longitudinal Bending in Buried Steel Pipes Subjected to Lateral Earth Movements by Mohamed Almahakeri, Ian D. Moore , and Amir Fam

Please find enclosed a revised manuscript with additional improvements to the presentation of the paper.

We'd like to thank you and all the reviewers for the time and efforts and look forward to seeing the paper in print.

Yours sincerely,

Mohamed Almahakeri